# Simple Hierarchical Planning with Diffusion

**Chang Chen[1], Fei Deng[1], Kenji Kawaguchi[2], Caglar Gulcehre[3,4]\*, Sungjin Ahn[5]†**

[1] Rutgers University, [2] National University of Singapore, [3] EPFL
[4] Google DeepMind, [5] KAIST

## Abstract

Diffusion-based generative methods have proven effective in modeling trajectories with offline datasets. However, they often face computational challenges and can falter in generalization, especially in capturing temporal abstractions for long-horizon tasks. To overcome this, we introduce the *Hierarchical Diffuser*, a simple, fast, yet surprisingly effective planning method combining the advantages of hierarchical and diffusion-based planning. Our model adopts a "jumpy" planning strategy at the higher level, which allows it to have a larger receptive field but at a lower computational cost—a crucial factor for diffusion-based planning methods, as we have empirically verified. Additionally, the jumpy sub-goals guide our low-level planner, facilitating a fine-tuning stage and further improving our approach's effectiveness. We conducted empirical evaluations on standard offline reinforcement learning benchmarks, demonstrating our method's superior performance and efficiency in terms of training and planning speed compared to the non-hierarchical Diffuser as well as other hierarchical planning methods. Moreover, we explore our model's generalization capability, particularly on how our method improves generalization capabilities on compositional out-of-distribution tasks.

## 1 Introduction

Planning has been successful in control tasks where the dynamics of the environment are known (Sutton & Barto, 2018; Silver et al., 2016). Through planning, the agent can simulate numerous action sequences and assess potential outcomes without interacting with the environment, which can be costly and risky. When the environment dynamics are unknown, a world model (Ha & Schmidhuber, 2018; Hafner et al., 2018; 2019) can be learned to approximate the true dynamics. Planning then takes place within the world model by generating future predictions based on actions. This type of model-based planning is considered more data-efficient than model-free methods and tends to transfer well to other tasks in the same environment (Moerland et al., 2023; Hamrick et al., 2020).

For temporally extended tasks with sparse rewards, the planning horizon should be increased accordingly (Nachum et al., 2019; Vezhnevets et al., 2017b; Hafner et al., 2022). However, this may not be practical as it requires an exponentially larger number of samples of action sequences to cover all possible plans adequately. Gradient-based trajectory optimization addresses this issue but can encounter credit assignment problems. A promising solution is to use hierarchical planning (Singh, 1992; Pertsch et al., 2020; Sacerdoti, 1974; Knoblock, 1990), where a high-level plan selects subgoals that are several steps apart, and low-level plans determine actions to move from one subgoal to the next. Both the high-level plan and each of the low-level plans are shorter than the original flat plan, leading to more efficient sampling and gradient propagation.

Conventional model-based planning typically involves separate world models and planners. However, the learned reward model can be prone to hallucinations, making it easy for the planner to exploit it (Talvitie, 2014). Recently, Janner et al. (2022b) proposed *Diffuser*, a framework where a single diffusion probabilistic model (Sohl-Dickstein et al., 2015; Ho et al., 2020; Song et al., 2021) is learned to serve as both the world model and the planner. It generates the states and actions in the full plan in parallel through iterative refinement, thereby achieving better global coherence. Furthermore, it also

---

\*This work was partly done while C.G. was at Google DeepMind. C.G. is currently affiliated with EPFL.
†Correspondence to `sungjin.ahn@kaist.ac.kr`

allows leveraging the guided sampling strategy Dhariwal & Nichol (2021) to provide the flexibility of adapting to the objective of the downstream task at test time.

Despite such advantages of Diffuser, how to enable hierarchical planning in the diffusion-based approach remains elusive to benefit from both diffusion-based and hierarchical planning simultaneously. Lacking this ability, Diffuser is computationally expensive and sampling inefficient due to the current dense and flat planning scheme. Moreover, we empirically found that the planned trajectories produced by Diffuser have inadequate coverage of the dataset distribution. This deficiency is particularly detrimental to diffusion-based planning.

In this paper, we propose the *Hierarchical Diffuser*, a simple framework that enables hierarchical planning using diffusion models. The proposed model consists of two diffusers: one for high-level subgoal generation and another for low-level subgoal achievement. To implement this framework, we first split each training trajectory into segments of equal length and consider the segment's split points as subgoals. We then train the two diffusers *simultaneously*. The high-level diffuser is trained on the trajectories consisting of only subgoals, which allows for a "jumpy" subgoal planning strategy and a larger receptive field at a lower computational cost. This sparseness reduces the diffusion model's burden of learning and sampling from high-dimensional distributions of dense trajectories, making learning and sampling more efficient. The low-level diffuser is trained to model only the segments, making it the subgoal achiever and facilitating a fine-tuning stage that further improves our approach's effectiveness. At test time, the high-level diffuser plans the jumpy subgoals first, and then the low-level diffuser achieves each subgoal by planning actions.

The contributions of this work are as follows. First, we introduce a diffusion-based hierarchical planning framework for decision-making problems. Second, we demonstrate the effectiveness of our approach through superior performance compared to previous methods on standard offline-RL benchmarks, as well as efficient training and planning speed. For example, our proposed method outperforms the baseline by 12.0% on Maze2D tasks and 9.2% on MuJoCo locomotion tasks. Furthermore, we empirically identify a key factor influencing the performance of diffusion-based planning methods, and showcase our method's enhanced generalization capabilities on compositional out-of-distribution tasks. Lastly, we provide a theoretical analysis of the generalization performance.

## 2 PRELIMINARIES

### 2.1 DIFFUSION PROBABILISTIC MODELS

Diffusion probabilistic models (Sohl-Dickstein et al., 2015; Ho et al., 2020; Song et al., 2021) have achieved state-of-the-art generation quality on various image generation tasks (Dhariwal & Nichol, 2021; Rombach et al., 2022; Ramesh et al., 2022; Saharia et al., 2022). They model the data generative process as $M$ steps of iterative denoising, starting from a Gaussian noise $\mathbf{x}_M \sim \mathcal{N}(\mathbf{0}, \mathbf{I})$:

$$p_\theta(\mathbf{x}_0) = \int p(\mathbf{x}_M) \prod_{m=0}^{M-1} p_\theta(\mathbf{x}_m \mid \mathbf{x}_{m+1}) \, \mathrm{d}\mathbf{x}_{1:M} \ . \tag{1}$$

Here, $\mathbf{x}_{1:M}$ are latent variables of the same dimensionality as the data $\mathbf{x}_0$, and

$$p_\theta(\mathbf{x}_m \mid \mathbf{x}_{m+1}) = \mathcal{N}(\mathbf{x}_m; \boldsymbol{\mu}_\theta(\mathbf{x}_{m+1}), \sigma_m^2 \mathbf{I}) \tag{2}$$

is commonly a Gaussian distribution with learnable mean and fixed covariance. The posterior of the latents is given by a predefined diffusion process that gradually adds Gaussian noise to the data:

$$q(\mathbf{x}_m \mid \mathbf{x}_0) = \mathcal{N}(\mathbf{x}_m; \sqrt{\bar{\alpha}_m}\mathbf{x}_0, (1 - \bar{\alpha}_m)\mathbf{I}) \ , \tag{3}$$

where the predefined $\bar{\alpha}_m \to 0$ as $m \to \infty$, making $q(\mathbf{x}_M \mid \mathbf{x}_0) \approx \mathcal{N}(\mathbf{0}, \mathbf{I})$ for a sufficiently large $M$.

In practice, the learnable mean $\boldsymbol{\mu}_\theta(\mathbf{x}_m)$ is often parameterized as a linear combination of the latent $\mathbf{x}_m$ and the output of a noise-prediction U-Net $\boldsymbol{\epsilon}_\theta(\mathbf{x}_m)$ (Ronneberger et al., 2015). The training objective is simply to make $\boldsymbol{\epsilon}_\theta(\mathbf{x}_m)$ predict the noise $\boldsymbol{\epsilon}$ that was used to corrupt $\mathbf{x}_0$ into $\mathbf{x}_m$:

$$\mathcal{L}(\theta) = \mathbb{E}_{\mathbf{x}_0, m, \boldsymbol{\epsilon}} \left[ \|\boldsymbol{\epsilon} - \boldsymbol{\epsilon}_\theta(\mathbf{x}_m)\|^2 \right] \ , \tag{4}$$

where $\mathbf{x}_m = \sqrt{\bar{\alpha}_m}\mathbf{x}_0 + \sqrt{1 - \bar{\alpha}_m}\boldsymbol{\epsilon}, \boldsymbol{\epsilon} \sim \mathcal{N}(\mathbf{0}, \mathbf{I})$.

## 2.2 Diffuser: Planning with Diffusion

Diffuser (Janner et al., 2022b) is a pioneering model for learning a diffusion-based planner from offline trajectory data. It has shown superior long-horizon planning capability and test-time flexibility. The key idea is to format the trajectories of states and actions into a two-dimensional array, where each column consists of the state-action pair at a single timestep:

$$\mathbf{x} = \begin{bmatrix} \mathbf{s}_0 & \mathbf{s}_1 & \dots & \mathbf{s}_T \\ \mathbf{a}_0 & \mathbf{a}_1 & \dots & \mathbf{a}_T \end{bmatrix} . \tag{5}$$

Diffuser then trains a diffusion probabilistic model $p_\theta(\mathbf{x})$ from an offline dataset. After training, $p_\theta(\mathbf{x})$ is able to jointly generate plausible state and action trajectories through iterative denoising. Importantly, $p_\theta(\mathbf{x})$ does not model the reward, and therefore is task-agnostic. To employ $p_\theta(\mathbf{x})$ to do planning for a specific task, Diffuser trains a separate guidance function $\mathcal{J}_\phi(\mathbf{x})$, and samples the planned trajectories from a perturbed distribution:

$$\tilde{p}_\theta(\mathbf{x}) \propto p_\theta(\mathbf{x}) \exp\left(\mathcal{J}_\phi(\mathbf{x})\right) . \tag{6}$$

Typically, $\mathcal{J}_\phi(\mathbf{x})$ estimates the expected return of the trajectory, so that the planned trajectories will be biased toward those that are plausible and also have high returns. In practice, $\mathcal{J}_\phi(\mathbf{x})$ is implemented as a regression network trained to predict the return $R(\mathbf{x})$ of the original trajectory $\mathbf{x}$ given a noise-corrupted trajectory $\mathbf{x}_m$ as input:

$$\mathcal{L}(\phi) = \mathbb{E}_{\mathbf{x},m,\boldsymbol{\epsilon}} \left[ \|R(\mathbf{x}) - \mathcal{J}_\phi(\mathbf{x}_m)\|^2 \right] , \tag{7}$$

where $R(\mathbf{x})$ can be obtained from the offline dataset, $\mathbf{x}_m = \sqrt{\bar{\alpha}_m}\mathbf{x} + \sqrt{1 - \bar{\alpha}_m}\boldsymbol{\epsilon}$, $\boldsymbol{\epsilon} \sim \mathcal{N}(\mathbf{0}, \mathbf{I})$.

Sampling from $\tilde{p}_\theta(\mathbf{x})$ is achieved similarly as classifier guidance (Dhariwal & Nichol, 2021; Sohl-Dickstein et al., 2015), where the gradient $\nabla_{\mathbf{x}_m}\mathcal{J}_\phi$ is used to guide the denoising process (Equation 2) by modifying the mean from $\boldsymbol{\mu}_m$ to $\tilde{\boldsymbol{\mu}}_m$:

$$\boldsymbol{\mu}_m \leftarrow \boldsymbol{\mu}_\theta(\mathbf{x}_{m+1}), \quad \tilde{\boldsymbol{\mu}}_m \leftarrow \boldsymbol{\mu}_m + \omega\sigma_m^2 \nabla_{\mathbf{x}_m}\mathcal{J}_\phi(\mathbf{x}_m)|_{\mathbf{x}_m = \boldsymbol{\mu}_m} . \tag{8}$$

Here, $\omega$ is a hyperparameter that controls the scaling of the gradient. To ensure that the planning trajectory starts from the current state $\mathbf{s}$, Diffuser sets $\mathbf{s}_0 = \mathbf{s}$ in each $\mathbf{x}_m$ during the denoising process. After sampling a full trajectory, Diffuser executes the first action in the environment, and replans at the next state $\mathbf{s}'$. In simple environments where replanning is unnecessary, the planned action sequence can be directly executed.

## 3 Hierarchical Diffuser

While Diffuser has demonstrated competence in long-horizon planning and test-time flexibility, we have empirically observed that its planned trajectories inadequately cover the dataset distribution, potentially missing high-return trajectories. Besides, the dense and flat planning scheme of the standard Diffuser is computationally expensive, especially when the planning horizon is long. Our key observation is that hierarchical planning could be an effective way to address these issues. To achieve this, we propose Hierarchical Diffuser, a simple yet effective framework that enables hierarchical planning while maintaining the benefits of diffusion-based planning. As shown in Figure 1, it consists of two Diffusers: one for high-level subgoal generation (Section 3.1) and the other for low-level subgoal achievement (Section 3.2).

## 3.1 Sparse Diffuser for Subgoal Generation

To perform hierarchical planning, the high-level planner needs to generate a sequence of intermediate states $(\mathbf{g}_1, \dots, \mathbf{g}_H)$ that serve as subgoals for the low-level planner to achieve. Here, $H$ denotes the planning horizon. Instead of involving complicated procedures for finding high-quality subgoals (Li et al., 2023) or skills (Rajeswar et al., 2023; Laskin et al., 2021), we opt for a simple approach that repurposes Diffuser for subgoal generation with minimal modification. In essence, we define the subgoals to be every $K$-th states and model the distribution of subsampled trajectories:

$$\mathbf{x}^{\mathrm{SD}} = \begin{bmatrix} \mathbf{s}_0 & \mathbf{s}_K & \dots & \mathbf{s}_{HK} \\ \mathbf{a}_0 & \mathbf{a}_K & \dots & \mathbf{a}_{HK} \end{bmatrix} =: \begin{bmatrix} \mathbf{g}_0 & \mathbf{g}_1 & \dots & \mathbf{g}_H \\ \mathbf{a}_0 & \mathbf{a}_K & \dots & \mathbf{a}_{HK} \end{bmatrix} . \tag{9}$$

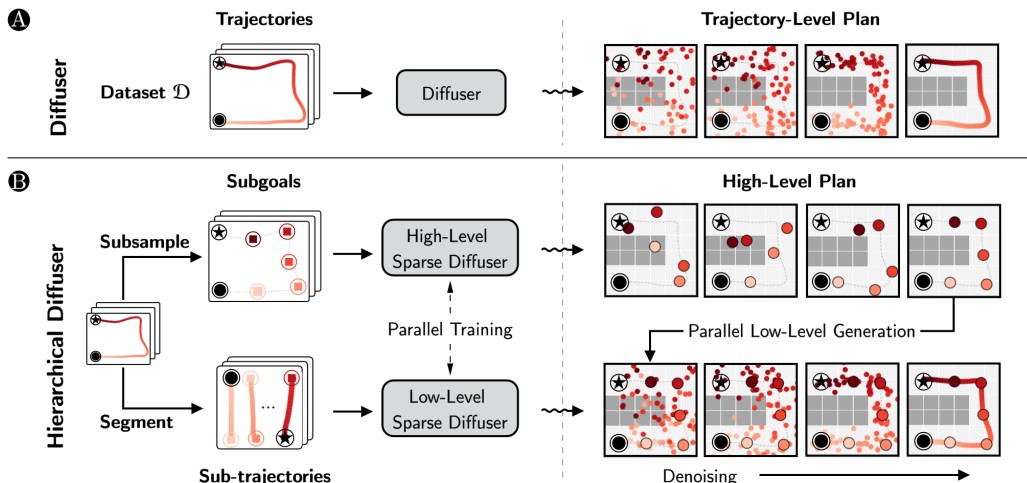

**Figure 1: Test and train-time differences between Diffuser models.** Hierarchical Diffuser (HD) is a general hierarchical diffusion-based planning framework. Unlike the Diffuser's training process (**A, left**), the HD's training phase reorganizes the training trajectory into two components: a sub-goal trajectory and dense segments. These components are then utilized to train the high-level and low-level denoising networks in parallel (**B, left**). During the testing phase, in contrast to Diffuser (**A, right**), HD initially generates a high-level plan consisted of sub-goals, which is subsequently refined through the low-level planner (**B, right**).

We name the resulting model *Sparse Diffuser (SD)*. While using every $K$-th states as subgoalas is a simplifying assumption, it is widely adopted in hierarchical RL due to its practical effectiveness (Zhang et al., 2023; Hafner et al., 2022; Li et al., 2022; Mandlekar et al., 2020; Vezhnevets et al., 2017a). We will empirically show that, desipite this simplicity, our approach is effective and efficient in practice, substantially outperforming HDMI (Li et al., 2023), a state-of-the-art method that adaptively selects subgoals.

The training procedure of Sparse Diffuser is almost the same as Diffuser. The only difference is that we need to provide the subsampled data $\mathbf{x}^{\mathrm{SD}}$ to the diffusion probabilistic model $p_{\theta_{\mathrm{SD}}}(\mathbf{x}^{\mathrm{SD}})$ and the guidance function $\mathcal{J}_{\phi_{\mathrm{SD}}}(\mathbf{x}^{\mathrm{SD}})$. It is important to note that, although the guidance function uses the subsampled data as input, it is still trained to predict the return of the full trajectory. Therefore, its gradient $\nabla_{\mathbf{x}^{\mathrm{SD}}}\mathcal{J}_{\phi_{\mathrm{SD}}}$ will direct toward a subgoal sequence that is part of high-return trajectories. However, due to the missing states and actions, the return prediction may become less accurate than Diffuser. In all of our experiments, we found that even if this is the case, it does not adversely affect task performance when compared to Diffuser. Moreover, our investigation suggests that including dense actions in $\mathbf{x}^{\mathrm{SD}}$ can improve return prediction and, in some environments, further improve task performance. We provide a detailed description in Section Section 3.4 and an ablation study in Section 4.3.

It is worth noting that Sparse Diffuser can itself serve as a standalone planner, without the need to involve any low-level planner. This is because Sparse Diffuser can generate the first action $\mathbf{a}_0$ of the plan, which is sufficient if we replan at each step. Interestingly, Sparse Diffuser already greatly outperforms Diffuser, mainly due to its increased receptive field (Section 4.3). While the receptive field of Diffuser can also be increased, this comes with hurting generalization performance and efficiency due to the increased model size (Appendix E and H).

## 3.2 FROM SPARSE DIFFUSER TO HIERARCHICAL DIFFUSER

While Sparse Diffuser can be used as a standalone planner, it only models the environment dynamics at a coarse level. This is beneficial for generating a high-level plan of subgoals, but it is likely that some low-level details are not taken into consideration. Therefore, we use a low-level planner to further refine the high-level plan, carving out the optimal dense trajectories that go from one subgoal to the next. This also allows us to avoid per-step replanning when it is not necessary. We call this two-level model *Hierarchical Diffuser (HD)*.

**Low-level Planner.** The low-level planner is simply implemented as a Diffuser $p_\theta(\mathbf{x}^{(i)})$ trained on trajectory segments $\mathbf{x}^{(i)}$ between each pair of adjacent subgoals $\mathbf{g}_i = \mathbf{s}_{iK}$ and $\mathbf{g}_{i+1} = \mathbf{s}_{(i+1)K}$:

$$\mathbf{x}^{(i)} = \begin{bmatrix} \mathbf{s}_{iK} & \mathbf{s}_{iK+1} & \cdots & \mathbf{s}_{(i+1)K} \\ \mathbf{a}_{iK} & \mathbf{a}_{iK+1} & \cdots & \mathbf{a}_{(i+1)K} \end{bmatrix} , \quad 0 \le i < H . \tag{10}$$

We also train a low-level guidance function $\mathcal{J}_\phi(\mathbf{x}^{(i)})$ that predicts the return $R(\mathbf{x}^{(i)})$ for each segment. The low-level Diffuser and guidance function are both shared across all trajectory segments, and they can be trained in parallel with the high-level planner.

**Hierarchical Planning.** After training the high-level and low-level planners, we use them to perform hierarchical planning as follows. Given a starting state $\mathbf{g}_0$, we first use the high-level planner to generate subgoals $\mathbf{g}_{1:H}$. This can be achieved by sampling from the perturbed distribution $\tilde{p}_{\theta_{\mathrm{SD}}}(\mathbf{x}^{\mathrm{SD}}) \propto p_{\theta_{\mathrm{SD}}}(\mathbf{x}^{\mathrm{SD}}) \exp\left(\mathcal{J}_{\phi_{\mathrm{SD}}}(\mathbf{x}^{\mathrm{SD}})\right)$, and then discarding the actions. Since the actions generated by the high-level planner are not used anywhere, in practice we remove the actions from subsampled trajectories $\mathbf{x}^{\mathrm{SD}}$ when training the high-level planner. In other words, we redefine

$$\mathbf{x}^{\mathrm{SD}} = \begin{bmatrix} \mathbf{s}_0 & \mathbf{s}_K & \cdots & \mathbf{s}_{HK} \end{bmatrix} =: \begin{bmatrix} \mathbf{g}_0 & \mathbf{g}_1 & \cdots & \mathbf{g}_H \end{bmatrix} . \tag{11}$$

Next, for each pair of adjacent subgoals $\mathbf{g}_i$ and $\mathbf{g}_{i+1}$, we use the low-level planner to generate a dense trajectory that connects them, by sampling from the distribution $\tilde{p}_\theta(\mathbf{x}^{(i)}) \propto p_\theta(\mathbf{x}^{(i)}) \exp\left(\mathcal{J}_\phi(\mathbf{x}^{(i)})\right)$. To ensure that the generated $\mathbf{x}^{(i)}$ indeed has $\mathbf{g}_i$ and $\mathbf{g}_{i+1}$ as its endpoints, we set $\mathbf{s}_{iK} = \mathbf{g}_i$ and $\mathbf{s}_{(i+1)K} = \mathbf{g}_{i+1}$ in each denoising step during sampling. Importantly, all low-level plans $\{\mathbf{x}^{(i)}\}_{i=0}^{H-1}$ can be generated in parallel. In environments that require per-step replanning, we only need to sample $\mathbf{x}^{(0)} \sim \tilde{p}_\theta(\mathbf{x}^{(0)})$, then execute the first action $\mathbf{a}_0$ in the environment, and replan at the next state. We highlight the interaction between the high-level and low-level planners in Appendix B.

### 3.3 IMPROVING RETURN PREDICTION WITH DENSE ACTIONS

**Sparse Diffuser with Dense Actions (SD-DA).** The missing states and actions in the subsampled trajectories $\mathbf{x}^{\mathrm{SD}}$ might pose difficulties in accurately predicting returns in certain cases. Therefore, we investigate a potential model improvement that subsamples trajectories with sparse states and dense actions. The hypothesis is that the dense actions can implicitly provide information about what has occurred in the intermediate states, thereby facilitating return prediction. Meanwhile, the sparse states preserve the model's ability to generate subgoals. We format the sparse states and dense actions into the following two-dimensional array structure:

$$\mathbf{x}^{\mathrm{SD\text{-}DA}} = \begin{bmatrix} \mathbf{s}_0 & \mathbf{s}_K & \cdots & \mathbf{s}_{HK} \\ \mathbf{a}_0 & \mathbf{a}_K & \cdots & \mathbf{a}_{HK} \\ \mathbf{a}_1 & \mathbf{a}_{K+1} & \cdots & \mathbf{a}_{HK+1} \\ \vdots & \vdots & \ddots & \vdots \\ \mathbf{a}_{K-1} & \mathbf{a}_{2K-1} & \cdots & \mathbf{a}_{(H+1)K-1} \end{bmatrix} =: \begin{bmatrix} \mathbf{g}_0 & \mathbf{g}_1 & \cdots & \mathbf{g}_H \\ \mathbf{a}_0 & \mathbf{a}_K & \cdots & \mathbf{a}_{HK} \\ \mathbf{a}_1 & \mathbf{a}_{K+1} & \cdots & \mathbf{a}_{HK+1} \\ \vdots & \vdots & \ddots & \vdots \\ \mathbf{a}_{K-1} & \mathbf{a}_{2K-1} & \cdots & \mathbf{a}_{(H+1)K-1} \end{bmatrix} , \tag{12}$$

where $\mathbf{a}_{\ge HK}$ in the last column are included for padding. Training proceeds similarly as Sparse Diffuser, where we train a diffusion model $p_{\theta_{\mathrm{SD\text{-}DA}}}(\mathbf{x}^{\mathrm{SD\text{-}DA}})$ to capture the distribution of $\mathbf{x}^{\mathrm{SD\text{-}DA}}$ in the offline dataset and a guidance function $\mathcal{J}_{\phi_{\mathrm{SD\text{-}DA}}}(\mathbf{x}^{\mathrm{SD\text{-}DA}})$ to predict the return of the full trajectory.

**Hierarchical Diffuser with Dense Actions (HD-DA).** This is obtained by replacing the high-level planner in Hierarchical Diffuser with SD-DA. The subgoals are generated by sampling from $\tilde{p}_{\theta_{\mathrm{SD\text{-}DA}}}(\mathbf{x}^{\mathrm{SD\text{-}DA}}) \propto p_{\theta_{\mathrm{SD\text{-}DA}}}(\mathbf{x}^{\mathrm{SD\text{-}DA}}) \exp\left(\mathcal{J}_{\phi_{\mathrm{SD\text{-}DA}}}(\mathbf{x}^{\mathrm{SD\text{-}DA}})\right)$, and then discarding the actions.

### 3.4 THEORETIC ANALYSIS

Theorem 1 in Appendix H demonstrates that the proposed method can improve the generalization capability of the baseline. Moreover, our analysis also sheds light on the tradeoffs in the value of $K$ and the kernel size. With a larger value of $K$, it is expected to have a better generalization gap for the diffusion process but a more loss of state-action details to perform RL tasks. With a larger kernel size, we expect a worse generalization gap for the diffusion process but a better receptive field to perform RL tasks. See Appendix H for more details.

**Table 1: Long-horizon Planning.** HD combines the benefits of both hierarchical and diffusion-based planning, achieving the best performance across all tasks. HD results are averaged over 100 planning seeds.

| Environment | | Flat Learning Methods | | | Hierarchical Learning Methods | | | |
|---|---|---|---|---|---|---|---|---|
| | | MPPI | IQL | Diffuser | IRIS | HiGoC | HDMI | HD (Ours) |
| Maze2D | U-Maze | 33.2 | 47.4 | 113.9±3.1 | - | - | 120.1±2.5 | **128.4**±3.6 |
| Maze2D | Medium | 10.2 | 34.9 | 121.5±2.7 | - | - | 121.8±1.6 | **135.6**±3.0 |
| Maze2D | Large | 5.1 | 58.6 | 123.0±6.4 | - | - | 128.6±2.9 | **155.8**±2.5 |
| **Single-task Average** | | 16.2 | 47.0 | 119.5 | - | - | 123.5 | **139.9** |
| Multi2D | U-Maze | 41.2 | 24.8 | 128.9±1.8 | - | - | 131.3±1.8 | **144.1**±1.2 |
| Multi2D | Medium | 15.4 | 12.1 | 127.2±3.4 | - | - | 131.6±1.9 | **140.2**±1.6 |
| Multi2D | Large | 8.0 | 13.9 | 132.1±5.8 | - | - | 135.4±2.5 | **165.5**±0.6 |
| **Multi-task Average** | | 21.5 | 16.9 | 129.4 | - | - | 132.8 | **149.9** |
| AntMaze | U-Maze | - | 62.2 | 76.0±7.6 | 89.4±2.4 | 91.2±1.9 | - | **94.0**±4.9 |
| AntMaze | Medium | - | 70.0 | 31.9±5.1 | 64.8±2.6 | 79.3±2.5 | - | **88.7**±8.1 |
| AntMaze | Large | - | 47.5 | 0.0±0.0 | 43.7±1.3 | 67.3±3.1 | - | **83.6**±5.8 |
| **AntMaze Average** | | - | 59.9 | 36.0 | 66.0 | 79.3 | - | **88.8** |

# 4 EXPERIMENTS

In our experiment section, we illustrate how and why the Hierarchical Diffuser (HD) improves Diffuser through hierarchcial planning. We start with our main results on the D4RL (Fu et al., 2020) benchmark. Subsequent sections provide an in-depth analysis, highlighting the benefits of a larger receptive field (RF) for diffusion-based planners for offline RL tasks. However, our compositional out-of-distribution (OOD) task reveals that, unlike HD, Diffuser struggles to augment its RF without compromising the generalization ability. Lastly, we report HD's efficiency in accelerating both the trainig time and planning time compared with Diffuser. The performance of HD across different $K$ values is detailed in the Appendix C. For the sake of reproducibility, we provide implementation and hyper-parameter details in Appendix A.

## 4.1 LONG-HORIZON PLANNING

We first highlight the advantage of hierarchical planning on long-horizon tasks. Specifically, we evaluate on Maze2D and AntMaze (Fu et al., 2020), two sparse-reward navigation tasks that can take hundreds of steps to accomplish. The agent will receive a reward of 1 when it reaches a fixed goal, and no reward elsewhere, making it challenging for even the best model-free algorithms (Janner et al., 2022b). The AntMaze adds to the challenge by having higher-dimensional state and action space. Following Diffuser (Janner et al., 2022b), we also evaluate multi-task flexibility on Multi2D, a variant of Maze2D that randomizes the goal for each episode.

**Results.** As shown in Table 1, Hierarchical Diffuser (HD) significantly outperforms previous state of the art across all tasks. The flat learning methods MPPI (Williams et al., 2016), IQL (Kostrikov et al., 2022), and Diffuser generally lag behind hierarchical learning methods, demonstrating the advantage of hierarchical planning. In addition, the failure of Diffuser in AntMaze-Large indicates that Diffuser struggles to simultaneously handle long-horizon planning and high-dimensional state and action space. Within hierarchical methods, HD outperforms the non-diffusion-based IRIS (Mandlekar et al., 2020) and HiGoC (Li et al., 2022), showing the benefit of planning with diffusion in the hierarchical setting. Compared with the diffusion-based HDMI (Li et al., 2023) that uses complex subgoal extraction procedures and more advanced model architectures, HD achieves >20% performance gain on Maze2D-Large and Multi2D-Large despite its simplicity.

## 4.2 OFFLINE REINFORCEMENT LEARNING

We further demonstrate that hierarchical planning generally improves offline reinforcement learning even with dense rewards and short horizons. We evaluate on Gym-MuJoCo and FrankaKitchen (Fu et al., 2020), which emphasize the ability to learn from data of varying quality and to generalize to unseen states, respectively. We use HD-DA as it outperforms HD in the dense reward setting. In addition to Diffuser and HDMI, we compare to leading methods in each task domain, including model-free BCQ (Fujimoto et al., 2019), BEAR (Kumar et al., 2019), CQL (Kumar et al., 2020), IQL (Kostrikov et al., 2022), Decision Transformer (DT; Chen et al., 2021), model-based MoReL (Ki-

Table 2: **Offline Reinforcement Learning.** HD-DA achieves the best overall performance. Results are averaged over 5 planning seeds. Following Kostrikov et al. (2022), we emphasize in bold scores within 5% of maximum.

| Gym Tasks | | BC | CQL | IQL | DT | TT | MOReL | Diffuser | HDMI | HD-DA (Ours) |
|---|---|---|---|---|---|---|---|---|---|---|
| Med-Expert | HalfCheetah | 55.2 | **91.6** | 86.7 | 86.8 | **95.0** | 53.3 | 88.9±0.3 | **92.1**±1.4 | **92.5**±0.3 |
| Med-Expert | Hopper | 52.5 | 105.4 | 91.5 | 107.6 | **110.0** | 108.7 | 103.3±1.3 | **113.5**±0.9 | **115.3**±1.1 |
| Med-Expert | Walker2d | **107.5** | **108.8** | **109.6** | **108.1** | 101.9 | 95.6 | 106.9±0.2 | **107.9**±1.2 | **107.1** ± 0.1 |
| Medium | HalfCheetah | 42.6 | 44.0 | **47.4** | 42.6 | **46.9** | 42.1 | 42.8 ± 0.3 | **48.0**±0.9 | **46.7**±0.2 |
| Medium | Hopper | 52.9 | 58.5 | 66.3 | 67.6 | 61.1 | **95.4** | 74.3 ± 1.4 | 76.4±2.6 | **99.3**±0.3 |
| Medium | Walker2d | 75.3 | 72.5 | 78.3 | 74.0 | 79.0 | 77.8 | 79.6±0.6 | **79.9**±1.8 | **84.0**±0.6 |
| Med-Replay | HalfCheetah | 36.6 | **45.5** | **44.2** | 36.6 | 41.9 | 40.2 | 37.7±0.5 | **44.9**±2.0 | 38.1±0.7 |
| Med-Replay | Hopper | 18.1 | **95.0** | **94.7** | 82.7 | 91.5 | 93.6 | 93.6±0.4 | **99.6**±1.5 | **94.7**±0.7 |
| Med-Replay | Walker2d | 26.0 | 77.2 | 73.9 | 66.6 | **82.6** | 49.8 | 70.6±1.6 | 80.7±2.1 | 84.1±2.2 |
| **Average** | | 51.9 | 77.6 | 77.0 | 74.7 | 78.9 | 72.9 | 77.5 | **82.6** | **84.6** |
| **Kitchen Tasks** | | BC | BCQ | BEAR | CQL | IQL | RvS-G | Diffuser | HDMI | HD-DA (Ours) |
| Partial | FrankaKitchen | 33.8 | 18.9 | 13.1 | 49.8 | 46.3 | 46.5 | 56.2 ± 5.4 | - | **73.3**±1.4 |
| Mixed | FrankaKitchen | 47.5 | 8.1 | 47.2 | 51.0 | 51.0 | 40.0 | 50.0 ± 8.8 | **69.2**±1.8 | **71.7**±2.7 |
| **Average** | | 40.7 | 13.5 | 30.2 | 50.4 | 48.7 | 43.3 | 53.1 | - | **72.5** |

Table 3: **Ablation on Model Variants.** SD yields an improvement over Diffuser, and the incorporation of low-level refinement in HD provides further enhancement in performance compared to SD.

| Dataset | Diffuser | SD | HD |
|---|---|---|---|
| Gym-MuJoCo | 77.5 | 80.7 | **81.7** |
| Maze2D | 119.5 | 133.4 | **139.9** |
| Multi2D | 129.4 | 145.8 | **149.9** |

Table 4: **Guidance Function Learning.** The included dense action helps learn guidance function, resulting in better RL performance.

| Dataset | $\mathcal{J}_\phi$ | | RL Performance | |
|---|---|---|---|---|
| | HD | HD-DA | HD | HD-DA |
| Hopper | 101.7 | **88.8** | 93.4±3.1 | **94.7**±0.7 |
| Walker2d | 166.1 | **133.0** | 77.2±3.3 | **84.1**±2.2 |
| HalfCheetah | 228.5 | **208.2** | 37.5±1.7 | **38.1**±0.7 |

dambi et al., 2020), Trajectory Transformer (TT; Janner et al., 2021), and Reinforcement Learning via Supervised Learning (RvS; Emmons et al., 2022).

**Results.** As shown in Table 2, HD-DA achieves the best average performance, significantly outperforming Diffuser while also surpassing the more complex HDMI. Notably, HD-DA obtains >35% improvement on FrankaKitchen over Diffuser, demonstrating its superior generalization ability.

## 4.3 ANALYSIS

To obtain a deeper understanding on HD improvements over Diffuser, we start our analysis with ablation studies on various model configurations. Insights from this analysis guide us to investigate the impact of effective receptive field on RL performance, specifically for diffusion-based planners. Furthermore, we introduce a compositional out-of-distribution (OOD) task to demonstrate HD's compositional generalization capabilities. We also evaluate HD's performance on varied jumpy step $K$ values to test its robustness and adaptability.

**SD already outperforms Diffuser. HD further improves SD via low-level refinement.** This can be seen from Table 3, where we report the performance of Diffuser, SD, and HD averaged over Maze2D, Multi2D, and Gym-MuJoCo tasks respectively. As mentioned in Section 3.1, here we use SD as a standalone planner. In the following, we investigate potential reasons why SD outperforms Diffuser.

**Large kernel size improves diffusion-based planning for in-distribution tasks.** A key difference between SD and Diffuser is that the subsampling in SD increases its effective receptive field. This leads us to hypothesize that a larger receptive field may be beneficial for modeling the data distribution, resulting in better performance. To test this hypothesis, we experiment with different kernel sizes of Diffuser, and report the averaged performance on Maze2D, Multi2D, and Gym-MuJoCo in Figure 2. We find that Diffuser's performance generally improves as the kernel size increases up to a certain threshold. (Critical drawbacks associated with increasing Diffuser's kernel sizes will be discussed in detail in the subsequent section.) Its best performance is comparable to SD, but remains inferior to HD. In Figure 3, we further provide a qualitative comparison of the model's coverage of the data distribution. We plot the actual executed trajectories when the agent follows the model-generated plans. Our results show that HD is able to generate plans that cover all distinct paths between the

start and goal state, exhibiting a distribution closely aligned with the dataset. Diffuser has a much worse coverage of the data distribution, but can be improved with a large kernel size.

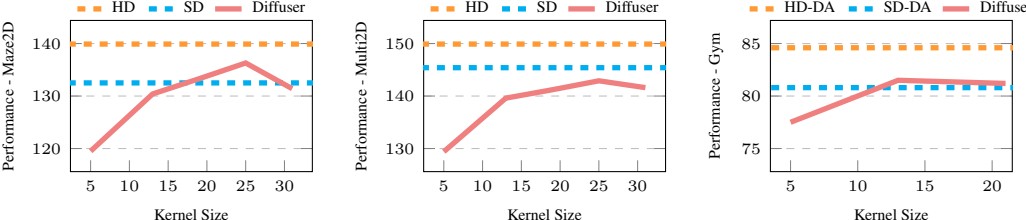

Figure 2: **Impact of Kernel Size.** Results of the impact of kernel size on performance of Diffuser in offline RL indicates that reasonably enlarging kernel size can improves the performance.

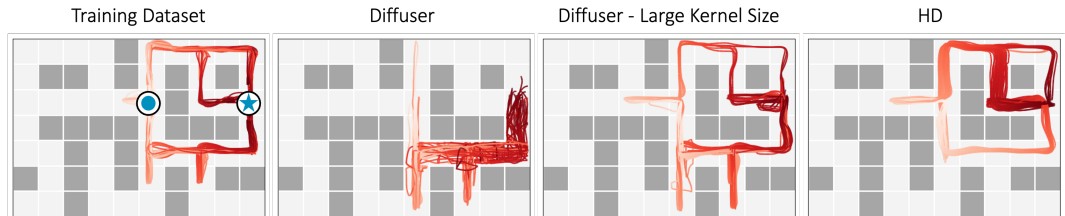

Figure 3: **Coverage of Data Distribution.** Empirically, we observed that Diffuser exhibits insufficient coverage of the dataset distribution. We illustrate this with an example featuring three distinct paths traversing from the start to the goal state. While Diffuser struggles to capture these divergent paths, both our method and Diffuser with an increased receptive field successfully recover this distribution.

**Large kernel size hurts out-of-distribution generalization.** While increasing the kernel size appears to be a simple way to improve Diffuser, it has many drawbacks such as higher memory consumption and slower training and planning. Most importantly, it introduces more model parameters, which can adversely affect the model's generalization capability. We demonstrate this in a task that requires the model to produce novel plans between unseen pairs of start and goal states at test time, by stitching together segments of training trajectories. We report the task success rate in Table 5, as well as the discrepancy between generated plans and optimal trajectories measured with cosine similarity and mean squared error (MSE). HD succeeds in all tasks, generating plans that are closest to the optimal trajectories, while Diffuser variants fail this task completely. Details can be found in Appendix E.

Table 5: **Out-Of-Distribution (OOD) Task Performance.** Only Hierarchical Diffuser (HD) can solve the compositional OOD task and generate plans that are most close to the optimal.

| Metrics | Diffuser-KS5 | Diffuser-KS13 | Diffuser-KS19 | Diffuser-KS25 | HD |
|---|---|---|---|---|---|
| Successful Rate | 0.0% | 0.0% | 0.0% | 0.0% | **100.0%** |
| Cosine Similarity | 0.85 | 0.89 | 0.93 | 0.93 | **0.98** |
| Deviation (MSE) | 1269.9 | 1311.1 | 758.5 | 1023.2 | **198.2** |

**Effect of Dense Actions.** Though the dense actions generated from high-level planer are discarded tn the low-level refinement phase, we empirically find that including dense actions facilitates the learning of the guidance function. As shown in Table 4, validation loss of guidance fuction learned from HD-DA is lower than that of SD-SA, leading to better RL performance. We conduct the experiment on the Medium-Replay dataset where learning the value function is hard due to the mixed policies.

**Efficiency Gains with Hierarchical Diffuser.** A potential concern when introducing an additional round of sampling might be the increase in planning time. However, the high-level plan, being $K$ times shorter, and the parallel generation of low-level segments counteract this concern. In Table 6, we observed a $10\times$ speed up over Diffuser in medium and large maze settings with horizons beyond 250 time steps. Details of the time measurement are in Appendix D.

Table 6: **Wall-clock Time Comparison.** Hierarchical Diffuser (HD) is more computationally efficient compared to Diffuser during both training and testing stages.

| Environment | Training [s] | | | | Planning [s] | | | |
|---|---|---|---|---|---|---|---|---|
| | U-Maze | Med-Maze | L-Maze | MuJoCo | U-Maze | Med-Maze | L-Maze | MuJoCo |
| HD | **8.0** | **8.7** | **8.6** | **9.9** | **0.8** | **3.1** | **3.3** | **1.0** |
| Diffuser | 26.6 | 132.7 | 119.7 | 12.3 | 1.1 | 9.9 | 9.9 | 1.3 |

## 5 Related Works

**Diffusion Models.** Diffusion models have recently emerged as a new type of generative model that supports generating samples, computing likelihood, and flexible-model complexity control. In diffusion models, the generation process is formulated as an iterative denoising process Sohl-Dickstein et al. (2015); Ho et al. (2020). The diffusion process can also be guided to a desired direction such as to a specific class by using either classifier-based guidance Nichol et al. (2021) or classifier-free guidance Ho & Salimans (2022). Recently, diffusion models have been adopted for agent learning. Janner et al. (2022b) have adopted it first and proposed the diffuser model which is the non-hierarchical version of our proposed model, while subsequent works by Ajay et al. (2022); Lu et al. (2023) optimized the guidance sampling process. Other works have utilized diffusion models specifically for RL Wang et al. (2022); Chen et al. (2023), observation-to-action imitation modeling Pearce et al. (2022), and for allowing equivariance with respect to the product of the spatial symmetry group Brehmer et al. (2023). A noteworthy contribution in this field is the hierarchical diffusion-based planning method Li et al. (2023), which resonates closely with our work but distinguishes itself in the subgoal preprocessing. While it necessitates explicit graph searching, our high-level diffuser to discover subgoals automatically.

**Hierarchical Planning.** Hierarchical planning has been successfully employed using temporal generative models, commonly referred to as world models Ha & Schmidhuber (2018); Hafner et al. (2019). These models forecast future states or observations based on historical states and actions. Recent years have seen the advent of hierarchical variations of these world models Chung et al. (2017); Kim et al. (2019); Saxena et al. (2021). Once trained, a world model can be used to train a separate policy with rollouts sampled from it Hafner et al. (2019); Deisenroth & Rasmussen (2011); Ghugare et al. (2023); Buckman et al. (2018); Hafner et al. (2022), or it can be leveraged for plan searching Schrittwieser et al. (2020); Wang & Ba (2020); Pertsch et al. (2020); Hu et al. (2023); Zhu et al. (2023). Our proposed method draws upon these principles, but also has connections to hierarchical skill-based planning such as latent skill planning Xie et al. (2020); Shi et al. (2022). However, a crucial distinction of our approach lies in the concurrent generation of all timesteps of a plan, unlike the aforementioned methods that require a sequential prediction of future states.

## 6 Conclusion

We introduce Hierarchical Diffuser, a comprehensive hierarchical framework that leverages the strengths of both hierarchical reinforcement learning and diffusion-based planning methods. Our approach, characterized by a larger receptive field at higher levels and a fine-tuning stage at the lower levels, has the capacity to not only capture optimal behavior from the offline dataset, but also retain the flexibility needed for compositional out-of-distribution (OOD) tasks. Expanding our methodology to the visual domain, which boasts a broader range of applications, constitutes another potential future direction.

**Limitations** Our Hierarchical Diffuser (HD) model has notable strengths but also presents some limitations. Foremost among these is its dependency on the quality of the dataset. Being an offline method, the performance of HD is restriced by the coverage or quality of datasets. In situations where it encounters unfamiliar trajectories, HD may struggle to produce optimal plans. Another restriction is the choice of fixed sub-goal intervals. This decision simplify the model's architecture but might fall short in handling a certain class of complex real-world scenarios. Furthermore, it introduces a task-dependent hyper-parameter. Lastly, the efficacy of HD is tied to the accuracy of the learned value function. This relationship places limits on the magnitude of the jump steps $K$; excessively skipping states poses challenge to learn the value function.

## ACKNOWLEDGEMENT

This work is supported by Brain Pool Plus Program (No. 2021H1D3A2A03103645) through the National Research Foundation of Korea (NRF) funded by the Ministry of Science and ICT. We would like to thank Michael Janner and Jindong Jiang for insightful discussions.

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

## A    IMPLEMENTATION DETAILS

In this section, we describe the details of implementation and hyperparameters we used during our experiments. For the Out-of-distribution experiment details, please check Section E.

- We build our Hierarchical Diffuser upon the officially released Diffuser code obtained from https://github.com/jannerm/diffuser. We list out the changes we made below.

- In our approach, the high-level and low-level planners are trained separately using segments randomly selected from the D4RL offline dataset.

- For the high-level planner's training, we choose segments equivalent in length to the planning horizon, $H$. Within these segments, states at every $K$ steps are selected. In the dense action variants, the intermediary action sequences between these states are then flattened concatenated with the corresponding jumpy states along the feature dimension. This approach of trajectory representation is also employed in the training of the high-level reward predictor.

- The sequence modeling at the low-level is the same as Diffuser except that we are using a sequence length of $K + 1$.

- We set $K = 15$ for the long-horizon planning tasks, while for the Gym-MuJoCo, we use $K = 4$.

- Aligning closely with the settings used by Diffuser, we employ a planning horizon of $H = 32$ for the MuJoCo locomotion tasks. For the Maze2D tasks, we utilize varying planning horizons; $H = 120$ for the Maze2D UMaze task, $H = 255$ for the Medium Maze task, and $H = 390$ for the Large Maze task. For the AntMaze tasks, we set $H = 225$ for the UMaze, $H = 255$ for the Medium Maze, and $H = 450$ for the Large Maze.

- For the MuJoCo locomotion tasks, we select the guidance scales $\omega$ from a set of choices, $\{0.1, 0.01, 0.001, 0.0001\}$, during the planning phase.

## B    PLANNING WITH HIGH-LEVEL DIFFUSER

We highlight the high-level planning and low-level planning in Algorithm 1 and Algorithm 2, respectively. The complete process of planning with HD is detailed in Algorithm 3

### B.1    PLANNING WITH HIGH-LEVEL DIFFUSER

The high-level module, Sparse Diffuser (SD), models the subsampled states and actions, enabling it to operate independently. We present the pseudocode of guided planning with the Sparse Diffuser in Algorithm 1.

### B.2    PLANNING WITH LOW-LEVEL DIFFUSER

Given subgoals sampled from the high-level diffuser, segments of low-level plans can be generated concurrently. We illatrate generating one such segment as example in Algorithm 2.

### B.3    HIERARCHICAL PLANNING

The comprehensive hierarchical planning involving both high-level and low-level planners is outlined in Algorithm 3. For the Maze2D tasks, we employed an open-loop approach, while for more challenging environments like AntMaze, Gym-MuJoCo, and Franka Kitchen, a closed-loop strategy was adopted.

---

**Algorithm 1** High-Level Planning

---

1: **function** SAMPLEHIGHLEVELPLAN(Current State $\mathbf{s}$, Sparse Diffuser $\boldsymbol{\mu}_{\theta_{\mathrm{SD}}}$, guidance function $\mathcal{J}_{\phi_{\mathrm{SD}}}$, guidance scale $\omega$, variance $\sigma_m^2$)
2:     initialize plan $\mathbf{x}_M^{\mathrm{SD}} \sim \mathcal{N}(\mathbf{0}, \mathbf{I})$
3:     **for** $m = M - 1, \ldots, 1$ **do**
4:         $\tilde{\boldsymbol{\mu}} \leftarrow \boldsymbol{\mu}_{\theta_{\mathrm{SD}}}(\mathbf{x}_{m+1}^{\mathrm{SD}}) + \omega\sigma_m^2 \nabla_{\mathbf{x}_m^{\mathrm{SD}}} \mathcal{J}_{\phi_{\mathrm{SD}}}(\mathbf{x}_m^{\mathrm{SD}})$
5:         $\mathbf{x}_{m-1}^{\mathrm{SD}} \sim \mathcal{N}(\tilde{\boldsymbol{\mu}}, \sigma_m^2 \mathbf{I})$
6:         Fix $\mathbf{g}_0$ in $\mathbf{x}_{m-1}^{\mathrm{SD}}$ to current state $\mathbf{s}$
7:     **end for**
8:     **return** High-level plan $\mathbf{x}_0^{\mathrm{SD}}$
9: **end function**

---

**Algorithm 2** Low-Level Planning

---

1: **function** SAMPLELOWLEVELPLAN(Subgoals $(g_i, g_{i+1})$, low-level diffuser $\boldsymbol{\mu}_\theta$, low-level guidance function $\mathcal{J}_\phi$, guidance scale $\omega$, variance $\sigma_m^2$)
2:     Initialize all low-level plan $\mathbf{x}_M^i \sim \mathcal{N}(\mathbf{0}, \mathbf{I})$
3:     **for** $m = M - 1, \ldots, 1$ **do**
4:         $\tilde{\boldsymbol{\mu}} \leftarrow \boldsymbol{\mu}_\theta(\mathbf{x}_{m+1}^i) + \omega\sigma_m^2 \nabla_{\mathbf{x}_m^i} \mathcal{J}_\phi(\mathbf{x}_m^i)$
5:         $\mathbf{x}_{m-1}^i \sim \mathcal{N}(\tilde{\boldsymbol{\mu}}, \sigma_m^2 \mathbf{I})$
6:         Fix $\mathbf{s}_0$ in $\mathbf{x}_{m-1}^i$ to $\mathbf{g}_i$; Fix $\mathbf{s}_K$ in $\mathbf{x}_{m-1}^i$ to $\mathbf{g}_{i+1}$
7:     **end for**
8:     **return** low-level plan $\mathbf{x}_0^i$
9: **end function**

---

## C ABLATION STUDY ON JUMPY STEPS K

In this section, we report the detailed findings from an ablation study concerning the impact of the parameter $K$ in Hierarchical Diffuser. The results, which are detailed in Tables 7 and 8, correspond to Maze2D tasks and MuJoCo locomotion tasks, respectively. As we increased $K$, an initial enhancement in performance was observed. However, a subsequent performance decline was noted with larger $K$ values. This trend aligns with our initial hypothesis that a larger $K$ introduces more skipped steps at the high-level planning stage, potentially resulting in the omission of information necessary for effective trajectory modeling, consequently leading to performance degradation.

**Table 7: Ablation on $K$ - Maze2D.** The model's performance increased with the value of $K$ up until $K = 21$. We report the mean and standard error over 100 random seeds.

| Environment | | K1 (Diffuser default) | HD-K7 | HD-K15 (default) | HD-K21 |
|---|---|---|---|---|---|
| Maze2D | U-Maze | $113.9 \pm 3.1$ | $127.0 \pm 1.5$ | $\mathbf{128.4 \pm 3.6}$ | $124.0 \pm 2.1$ |
| Maze2D | Medium | $121.5 \pm 2.7$ | $132.5 \pm 1.3$ | $\mathbf{135.6 \pm 3.0}$ | $130.3 \pm 2.4$ |
| Maze2D | Large | $123.0 \pm 6.4$ | $153.2 \pm 3.0$ | $\mathbf{155.8 \pm 2.5}$ | $158.9 \pm 2.0$ |
| **Sing-task Average** | | 119.5 | 137.6 | **139.9** | 137.7 |
| Multi2D | U-Maze | $128.9 \pm 1.8$ | $135.4 \pm 1.1$ | $\mathbf{144.1 \pm 1.2}$ | $133.7 \pm 1.3$ |
| Multi2D | Medium | $127.2 \pm 3.4$ | $135.3 \pm 1.6$ | $\mathbf{140.2 \pm 1.6}$ | $134.5 \pm 1.4$ |
| Multi2D | Large | $132.1 \pm 5.8$ | $160.2 \pm 1.9$ | $\mathbf{165.5 \pm 0.6}$ | $159.3 \pm 3.0$ |
| **Multi-task Average** | | 129.4 | 143.7 | **149.9** | 142.5 |

## D WALL CLOCK COMPARISON DETAILS

We evaluated the wall clock time by averaging the time taken per complete plan during testing and, for the training phase, the time needed for 100 updates. All models were measured using a single NVIDIA RTX 8000 GPU to ensure consistency. We employ the released code and default settings for the Diffuser model. We select the Maze2D tasks and Hopper-Medium-Expert, a representative for

---

**Algorithm 3** Hierarchical Planning

---

1: **function** SAMPLEHIERARCHICALPLAN(High-level diffuser $\boldsymbol{\mu}_{\theta_{\text{SD}}}$, low-level diffuser $\boldsymbol{\mu}_\theta$, high-level guidance function $\mathcal{J}_{\phi_{\text{SD}}}$, low-level guidance function $\mathcal{J}_\phi$, high-level guidance scale $\omega_{\text{SD}}$, low-level guidance scale $\omega$, high-level variance $\sigma^2_{\text{SD},m}$, low-level variance $\sigma^2_m$)
2:     Observe state $\mathbf{s}$;
3:     **if** do open-loop **then**
4:         Sample high-level plan $\mathbf{x}^{\text{SD}}$ = SAMPLEHIGHLEVELPLAN($\mathbf{s}$, $\boldsymbol{\mu}_{\theta_{\text{SD}}}$, $\mathcal{J}_{\phi_{\text{SD}}}$, $\omega_{\text{SD}}$, $\sigma^2_{SD,m}$)
5:         **for** $i = 0, \ldots, H - 1$ **parallel do**
6:             Sample low-level plan $\mathbf{x}^{(i)}$ = SAMPLELOWLEVELPLAN($(g_i, g_{i+1})$, $\boldsymbol{\mu}_\theta$, $\mathcal{J}_\phi$, $\omega$, $\sigma^2_m$)
7:         **end for**
8:         Form the full plan $\mathbf{x}$ with low-level plans $\mathbf{x}^{(i)}$ for $i = 0, H - 1$
9:         **for** action $\mathbf{a}_t$ in $\mathbf{x}$ **do**
10:             Execute $\mathbf{a}_t$
11:         **end for**
12:     **else**
13:         **while** not done **do**
14:             Sample high-level plan $\mathbf{x}^{\text{SD}}$ = SAMPLEHIGHLEVELPLAN($\mathbf{s}$, $\boldsymbol{\mu}_{\theta_{\text{SD}}}$, $\mathcal{J}_\phi$, $\omega_{\text{SD}}$, $\sigma^2_{\text{SD},m}$)
15:             *// Sample only the first low-level segment*
16:             Sample $\mathbf{x}^{(0)}$ = SAMPLELOWLEVELPLAN($(g_0, g_1)$, $\boldsymbol{\mu}_\theta$, $\mathcal{J}_\phi$, $\omega$, $\sigma^2_m$)
17:             Execute the first $\mathbf{a}_0$ of plan $\mathbf{x}^{(0)}$
18:             Observe state $\mathbf{s}$
19:         **end while**
20:     **end if**
21: **end function**

---

**Table 8: Ablation on $K$ - MuJoCo Locomotion.** The model's performance increased with the value of $K$ up until $K = 8$. We report the mean and standard error over 5 random seeds.

| Dataset | Environment | K1 (Diffuser default) | HD-K4 (default) | HD-K8 |
|---|---|---|---|---|
| Medium-Expert | HalfCheetah | $88.9 \pm 0.3$ | $\mathbf{92.5 \pm 0.3}$ | $91.5 \pm 0.3$ |
| Medium-Expert | Hopper | $103.3 \pm 1.3$ | $\mathbf{115.3 \pm 1.1}$ | $113.0 \pm 0.5$ |
| Medium-Expert | Walker2d | $106.9 \pm 0.2$ | $107.1 \pm 0.1$ | $\mathbf{107.6 \pm 0.3}$ |
| Medium | HalfCheetah | $42.8 \pm 0.3$ | $\mathbf{46.7 \pm 0.2}$ | $45.9 \pm 0.7$ |
| Medium | Hopper | $74.3 \pm 1.4$ | $\mathbf{99.3 \pm 0.3}$ | $86.7 \pm 7.4$ |
| Medium | Walker2d | $79.6 \pm 0.6$ | $84.0 \pm 0.6$ | $\mathbf{84.2 \pm 0.5}$ |
| Medium-Replay | HalfCheetah | $37.7 \pm 0.5$ | $38.1 \pm 0.7$ | $\mathbf{39.5 \pm 0.4}$ |
| Medium-Replay | Hopper | $93.6 \pm 0.4$ | $\mathbf{94.7 \pm 0.7}$ | $91.3 \pm 1.3$ |
| Medium-Replay | Walker2d | $70.6 \pm 1.6$ | $\mathbf{84.1 \pm 2.2}$ | $76.4 \pm 2.7$ |
| **Average** | | $77.5$ | $\mathbf{84.6}$ | $81.8$ |

the Gym-MuJoCo tasks, from the D4RL benchmark for our measurement purpose. On the Maze2D tasks, we set $K = 15$, and for the Gym-MuJoCo tasks, we set it to $4$ as this is our default setting for RL tasks. The planning horizons of HD for each task, outlined in Table 9, are influenced by their need for divisibility by $K$, leading to slight deviations from the default values used by the Diffuser.

**Table 9:** Wall-clock time $H$ value

| Environment | Diffuser | Ours |
|---|---|---|
| Maze2d-Umaze | 128 | 120 |
| Maze2d-Medium | 256 | 255 |
| Maze2d-Large | 384 | 390 |
| Hopper-Medium-Expert | 32 | 32 |

# E   COMPOSITIONAL OUT-OF-DISTRIBUTION (OOD) EXPERIMENT DETAILS

While an increase in kernel size does indeed provide a performance boost for the Diffuser model, this enlargement inevitably augments the model's capacity, which potentially increases the risk of overfitting. Therefore, Diffuser models may underperform on tasks demanding both a large receptive field and strong generalization abilities. To illustrate this, inspired by Janner et al. (2022a), we designed a compositional out-of-distribution (OOD) Maze2D task, as depicted in Figure 4. During training, the agent is only exposed to offline trajectories navigating diagonally. However, during testing, the agent is required to traverse between novel start-goal pairs. We visualized the 32 plans generated by the models in Figure 4. As presented in the figure, only the Hierarchical Diffuser can generate reasonable plans approximating the optimal solution. In contrast, all Diffuser variants either create plans that lead the agent crossing a wall (i.e. Diffuser, Diffuser-KS13, and Diffuser-KS19) or produce plans that exceed the maximum step limit (i.e. Diffuser-13, Diffuser-KS19, and Diffuser-KS25).

To conduct this experiment, we generated a training dataset of 2 million transitions using the same Proportional-Derivative (PD) controller as used for generating the Maze2D tasks. Given that an optimal path typically requires around 230 steps to transition from the starting point to the end goal, we set the planning horizon $H$ for the Diffuser variants at 248, while for our proposed method, we set it at 255, to ensure divisibility by $K = 15$. For the reinforcement learning task in the testing phase, the maximum steps allowed were set at 300. Throughout the training phase, we partitioned 10% of the training dataset as a validation set to mitigate the risk of overfitting. To quantitatively measure the discrepancy between the generated plans and the optimal solution, we used Cosine Similarity and Mean Squared Error (MSE). Specifically, we crafted 10 optimal paths using the same controller and sampled 100 plans from each model for each testing task. To ensure that the optimal path length aligned with the planning horizon of each model, we modified the threshold distance used to terminate the controller once the agent reached the goal state. Subsequently, we computed the discrepancy between each plan and each optimal path. The mean of these results was reported in Table 5.

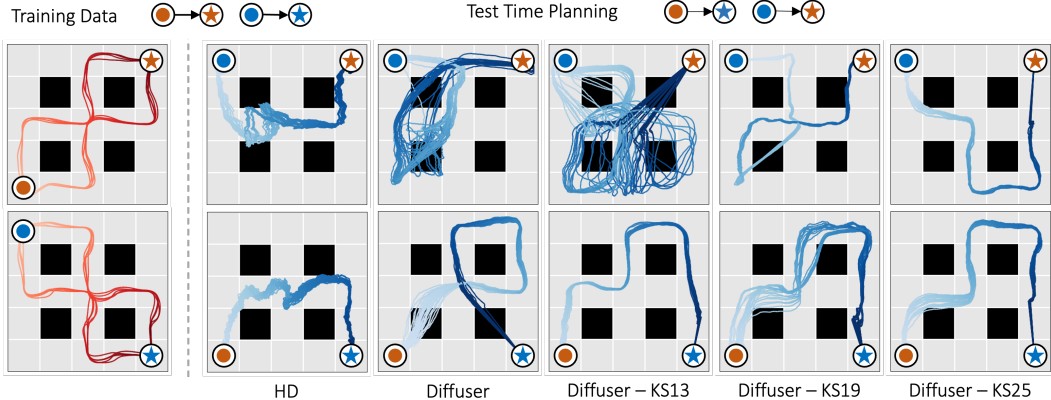

**Figure 4: Large Kernel Size Hurts the OOD Generalization.** Increasing kernel size generally improves the offline RL performance of Diffuser model. However, when a large receptive field and compositional out-of-distribution (OOD) generalization are both required, Diffuser models offer no simple solution. We demonstrate this with the sampled plans from both the standard Difuser and a Difuser with varied kernel sizes (KS). None of them can come up with an optimal plan by stiching training segments together. Conversely, our proposed Hierarchical Diffuser (HD) possesses both a large receptive field and the flexibility needed of compositional OOD tasks.

# F   OOD GENERALIZATION IN MOTION PLANNING

We carried out an additional experiment focused on out-of-distribution (OOD) scenarios, assessing the model's capability to navigate through unseen obstacles. Following the methodology of MPD

(Carvalho et al., 2023), we applied our HD model to the PointMass2D Dense task with OOD obstacles. To do this, we replaced the flat Diffuser planner used in MPD with our HD model, referred to as MP-HD. We used the official code of MPD for comparison.

Our HD model achieved a noteworthy success rate of $81.0 \pm 38.8$, surpassing MPD's performance of $75.0 \pm 43.3$. We also visualized sample trajectories of HD from two randomly selected (start, goal) pairs in Figure 5. These trajectories demonstrate that our model can effectively avoid collisions when faced with OOD test obstacles.

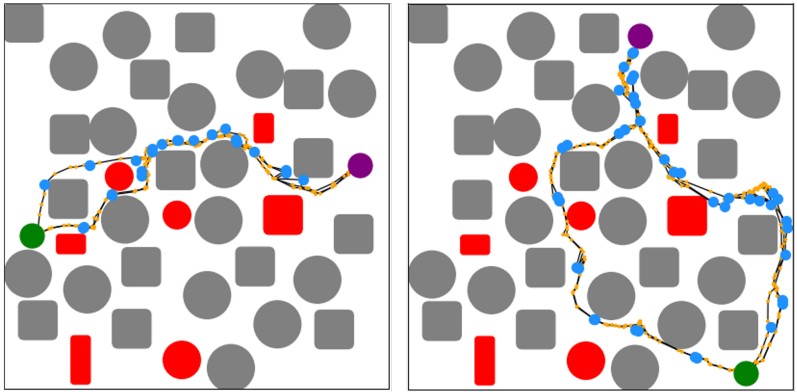

**Figure 5: Sample Trajectories with Unseen Obstacles.** HD generate multiple paths navigating from two randomly selected start and goal states. Obstacles in red were not present during training. We marked sub-goal states in cyan for clarity. We marked sub-goal states in cyan for clarity. The trajectories begin at the green circle, with the target state represented by a purple circle.

## G  ADDITIONAL ABLATION STUDIES

### G.1  TRANSFORMER-BASED DIFFUSION

We compare our model (based on U-Net (CNN)) with Transformer-based diffusion in this section. For this experiment, we use the hyperparameter setting in the Decision Transformer (Chen et al., 2021) as a starting point for our investigation. The results, as shown in the Table 10, reveal that the HD-Transformer achieves similar performance to the HD-UNet in Maze2D tasks, though it is slightly less effective in the Gym-MuJoCo tasks. While the HD-Transformer shows promise, we would like to emphasize that our primary contribution is not the backbone architecture but the benefits of hierarchical structures.

### G.2  SUB-GOAL SELECTION STRATEGIES

Hierarchical Diffuser (HD) select sub-goals with fixed time interval for simplicity. Here, we consider other choises:

- **Route Sampling** (Lai et al., 2020) (RS): In line with HDMI, we also consider choosing waypoint with fixed length interval as sub-goals. Specifically, denote the distance moved after action $a_t$ as $\delta_t$. Then, the route length can be computed as $S = \sum_{t=0}^{T1} \delta_t$ . We pick the waypoints with fixed interval of $S/k$, where $k$ is the number of sub-goals.

- **Value Sampling** (Correia & Alexandre, 2023) (VS): Also inspired by HDMI, we also test the value sampling method, where the most valuable states are chosen as sub-goals. Specifically, the distance weighted accumulated reward is used to value each states after state $s_i$: $W(s_j) = \sum_{k=i+1}^{j} \frac{r_k}{j-i}$.

- **Future Sampling** (Andrychowicz et al., 2017) (FS): Beyond RS and VS, we also explored a hindsight heuristic method, randomly selecting future states as sub-goals.

**Table 10: Ablation Study on Backbone Architecture.** HD-Transformer achieves comparable with HD-Unet on a wide rang of tasks. We report the mean and standard error over 5 random seeds.

| Task | HD-UNet | HD-Transformer |
|---|---|---|
| Maze2d-Large | $\mathbf{128.4} \pm 3.6$ | $127.9 \pm 3.2$ |
| Maze2d-Medium | $135.6 \pm 3.0$ | $\mathbf{136.1} \pm 2.6$ |
| Maze2d-UMaze | $\mathbf{155.8} \pm 2.5$ | $154.1 \pm 3.6$ |
| **Maze2d Average** | **139.9** | 139.4 |
| MedExp-HalfCheetah | $\mathbf{92.5} \pm 0.3$ | $88.4 \pm 0.6$ |
| MedExp-Hopper | $\mathbf{115.3} \pm 1.1$ | $103.9 \pm 5.9$ |
| MedExp-Walker2d | $\mathbf{107.1} \pm 0.1$ | $107.0 \pm 0.3$ |
| Medium-HalfCheetah | $\mathbf{46.7} \pm 0.2$ | $45.3 \pm 0.5$ |
| Medium-Hopper | $\mathbf{99.3} \pm 0.3$ | $94.0 \pm 5.4$ |
| Medium-Walker2d | $\mathbf{84.0} \pm 0.6$ | $82.8 \pm 1.7$ |
| MedRep-HalfCheetah | $38.1 \pm 0.7$ | $\mathbf{39.5} \pm 0.2$ |
| MedRep-Hopper | $\mathbf{94.7} \pm 0.7$ | $91.4 \pm 1.5$ |
| MedRep-Walker2d | $\mathbf{84.1} \pm 2.2$ | $81.2 \pm 1.1$ |
| **Gym Average** | **84.6** | 81.5 |

Notably, in RS and VS, certain states might never be chosen as sub-goals, unlike FS and the fixed time interval sampling (TS) used in HD, which offers equal probability for each state to be selected as a sub-goal.

Given the varying lengths of sub-tasks generated by these selection methods, integrating dense action at the high level was impractical. Hence, we focused our experiments on HD rather than HD-DA. At the low level, sub-trajectories were padded to a consistent length $L$. It's important to note that excluding dense action data at the high level may slightly hinder the learning of the value function, potentially leading to a marginal decrease in performance. The results, as presented in the Table 11, demonstrate that our hierarchical framework is generally resilient across different sub-goal selection methods. While HD-VS and HD-RS exhibited somewhat lower performance, we hypothesize this may be due to uneven sampling of valuable states, which could impact the planning guidance function's effectiveness.

**Table 11: Ablation Study on Sub-goal Selection.** HD is generally resilient across different sub-goal selection methods. We report the mean and standard error over 5 random seeds.

| Dataset | HD-DA | HD | HD-FS | HD-VS | HD-RS |
|---|---|---|---|---|---|
| MedExp-Halfcheetah | $\mathbf{92.5} \pm 0.3$ | $92.1 \pm 0.5$ | $87.6 \pm 0.7$ | $87.6 \pm 0.6$ | $88.4 \pm 0.4$ |
| MedExp-Hopper | $\mathbf{115.3} \pm 1.1$ | $104.1 \pm 8.2$ | $106.5 \pm 5.5$ | $108.9 \pm 4.8$ | $106.4 \pm 5.0$ |
| MedExp-Walker2d | $107.1 \pm 0.1$ | $\mathbf{107.4} \pm 0.3$ | $107.0 \pm 0.1$ | $\mathbf{107.4} \pm 0.2$ | $\mathbf{107.4} \pm 0.3$ |
| Medium-Halfcheetah | $\mathbf{46.7} \pm 0.2$ | $45.2 \pm 0.2$ | $43.9 \pm 0.4$ | $43.2 \pm 0.3$ | $43.6 \pm 0.9$ |
| Medium-Hopper | $99.3 \pm 0.3$ | $99.2 \pm 0.7$ | $\mathbf{100.9} \pm 0.8$ | $92.3 \pm 4.2$ | $95.8 \pm 1.3$ |
| Medium-Walker2d | $\mathbf{84.0} \pm 0.6$ | $82.6 \pm 0.8$ | $83.1 \pm 1.0$ | $82.4 \pm 0.9$ | $82.9 \pm 1.1$ |
| MedRep-Halfcheetah | $38.1 \pm 0.7$ | $37.5 \pm 1.7$ | $\mathbf{39.7} \pm 0.3$ | $38.1 \pm 0.7$ | $38.4 \pm 0.8$ |
| MedRep-Hopper | $\mathbf{94.7} \pm 0.7$ | $93.4 \pm 3.1$ | $90.9 \pm 1.7$ | $91.3 \pm 1.3$ | $92.6 \pm 1.2$ |
| MedRep-Walker2d | $\mathbf{84.1} \pm 2.2$ | $77.2 \pm 3.3$ | $80.9 \pm 1.7$ | $75.7 \pm 2.1$ | $76.4 \pm 2.7$ |
| **Average** | **84.6** | 82.1 | 82.3 | 80.8 | 81.3 |

## H  THEORETICAL ANALYSIS

In this section, we show that the proposed method can improve the generalization capability when compared to the baseline. Our analysis also sheds light on the tradeoffs in $K$ and the kernel size. Let $K \in \{1, \dots, T\}$, $\ell(x) = \tau \mathbb{E}_{m,\epsilon}[\|\epsilon - \epsilon_\theta(\sqrt{\bar{\alpha}_m}x + \sqrt{1 - \bar{\alpha}_m}\epsilon, m)\|^2]]$, where $\tau > 0$ is an arbitrary normalization coefficient that can depend on $K$: e.g., $1/d$ where $d$ is the dimensionality of $\epsilon$. Given the training trajectory data $(\mathbf{x}_0^{(i)})_{i=1}^n$, the training loss is defined by $\hat{\mathcal{L}}(\theta) = \frac{1}{n} \sum_{i=1}^n \ell(\mathbf{x}_0^{(i)})$ where

$\mathbf{x}_m^{(i)} = \sqrt{\bar{\alpha}_m}\mathbf{x}_0^{(i)} + \sqrt{1-\bar{\alpha}_m}\epsilon$, and $\mathbf{x}_0^{(1)},\ldots,\mathbf{x}_0^{(n)}$ are independent samples of trajectories. We have $\mathcal{L}(\theta) = \mathbb{E}_{\mathbf{x}_0}[\ell(\mathbf{x}_0)]$. Define $\hat{\theta}$ to be an output of the training process using $(\mathbf{x}_0^{(i)})_{i=1}^n$, and $\varphi$ to be the (unknown) value function under the optimal policy. Let $\Theta$ be the set of $\theta$ such that $\hat{\theta} \in \Theta$ and $\Theta$ is independent of $(\mathbf{x}_0^{(i)})_{i=1}^n$. Denote the projection of the parameter space $\Theta$ onto the loss function by $\mathcal{H} = \{x \mapsto \tau\mathbb{E}_{m,\epsilon}[\|\epsilon - \epsilon_\theta(\sqrt{\bar{\alpha}_m}x + \sqrt{1-\bar{\alpha}_m}\epsilon, m)\|^2] : \theta \in \Theta\}$, the conditional Rademacher complexity by $\mathcal{R}_t(\mathcal{H}) = \mathbb{E}_{(\mathbf{x}_0^{(i)})_{i=1}^n,\xi}[\sup_{h\in\mathcal{H}} \frac{1}{n_t}\sum_{i=1}^{n_t}\xi_i h(\mathbf{x}_0^{(i)}) \mid \mathbf{x}_0^{(i)} \in \mathcal{C}_t]$, where $\mathcal{C}_t = \{\mathbf{x}_0 \in \mathcal{X} : t = \arg\max_{j\in[H]}\varphi(\mathbf{g}_j)$ where $[\mathbf{g}_1\,\mathbf{g}_2\,\cdots\,\mathbf{g}_H]$ is the first row of $\mathbf{x}_0\}$ and $n_t = \sum_{i=1}^n \mathbb{1}\{\mathbf{x}_0^{(i)} \in \mathcal{C}_t\}$. Define $\mathcal{T} = \{t \in [H] : n_t \geq 1\}$ and $C_0 = d\tau c((1/\sqrt{2}) + \sqrt{2})$ for some $c \geq 0$ such that $c \geq \mathbb{E}_{m,\epsilon}[((\epsilon - \epsilon_\theta(\mathbf{x}_m, m))_i)^2]]$ for $i = 1,\ldots,d$, where $d$ is the dimension of $\epsilon \in \mathbb{R}^d$. Here, both the loss values and $C_0$ scale linearly in $d$. Our theorem works for any $\tau > 0$, including $\tau = 1/d$, which normalizes the loss values and $C_0$ with respect to $d$. Thus, the conclusion of our theorem is invariant of the scale of the loss value.

**Theorem 1.** *For any $\delta > 0$, with probability at least $1 - \delta$,*

$$\mathcal{L}(\hat{\theta}) \leq \hat{\mathcal{L}}(\hat{\theta}) + C_0\sqrt{\left\lceil\frac{T}{K}\right\rceil\frac{\ln(\lceil\frac{T}{K}\rceil\frac{2}{\delta})}{n}} + \sum_{t\in\mathcal{T}}\frac{2n_t\mathcal{R}_t(\mathcal{H})}{n}. \tag{13}$$

The proof is presented in Appendix J. The baseline is recovered by setting $K = 1$. Thus, Theorem 1 demonstrates that the proposed method (i.e., the case of $K > 1$) can improve the generalization capability of the baseline (i.e., the case of $K = 1$). Moreover, while the upper bound on $\mathcal{L}(\hat{\theta}) - \hat{\mathcal{L}}(\hat{\theta})$ decreases as $K$ increases, it is expected that we loose more details of states with a larger value of $K$. Therefore, there is a tradeoff in $K$: i.e., with a larger value of $K$, we expect a better generalization for the diffusion process but a more loss of state-action details to perform RL tasks. On the other hand, the conditional Rademacher complexity term $\mathcal{R}_t(\mathcal{H})$ in Theorem 1 tends to increase as the number of parameters increases. Thus, there is also a tradeoff in the kernel size: i.e., with a larger kernel size, we expect a worse generalization for the diffusion process but a better receptive field to perform RL tasks. We provide the additional analysis on $\mathcal{R}_t(\mathcal{H})$ in Appendix I.

## I ON THE CONDITIONAL RADEMACHER COMPLEXITY

In this section, we state that the term $\sum_{t\in\mathcal{T}}\frac{2n_t\mathcal{R}_t(\mathcal{H})}{n}$ in Theorem 1 is also smaller for the proposed method with $K \geq 2$ when compared to the base model (i.e., with $K = 1$) under the following assumptions that typically hold in practice. We assume that we can express $\epsilon_\theta(\mathbf{x}_m, m) = Wg(V\mathbf{x}_m, m)$ for some functions $g$ and some matrices $W, V$ such that the parameters of $g$ do not contain the entries of $W$ and $V$, and that $\Theta$ contains $\theta$ with $W$ and $V$ such that $\|W\|_\infty \leq \zeta_W$ and $\|V\|_\infty < \zeta_V$ for some $\zeta_W$ and $\zeta_V$. This assumption is satisfied in most neural networks used in practice as $g$ is arbitrarily; e.g., we can set $g = \epsilon_\theta$, $W = I$ and $V = I$ to have any arbitrary function $\epsilon_\theta(\mathbf{x}_m, m) = Wg(V\mathbf{x}_m, m) = g(\mathbf{x}_m, m)$. We also assume that $\mathcal{R}_t(\mathcal{H})$ does not increase when we increase $n_t$. This is reasonable since $\mathcal{R}_t(\mathcal{H}) = O(\frac{1}{n_t})$ for many machine learning models, including neural networks. Under this setting, the following proposition states that the term $\sum_{t\in\mathcal{T}}\frac{2n_t\mathcal{R}_t(\mathcal{H})}{n}$ of with the proposed method is also smaller than that of the base model:

**Proposition 1.** *Let $q \geq 2$ and denote by $\bar{\mathcal{R}}_t(\bar{\mathcal{H}})$ and $\tilde{\mathcal{R}}_t(\tilde{\mathcal{H}})$ the conditional Rademacher complexities for $K = 1$ (base case) and $K \geq q$ (proposed method) respectively. Then, $\bar{\mathcal{R}}_t(\bar{\mathcal{H}}) \geq \tilde{\mathcal{R}}_t(\tilde{\mathcal{H}})$ for any $t \in \{1,\ldots,T\}$ such that $s_t$ is not skipped with $K = q$.*

The proof is presented in Appendix J.

## J PROOFS

### J.1 PROOF OF THEOREM 1

*Proof.* Let $K \in \{1,\ldots,T\}$. Define $[H] = \{1,\ldots,H\}$. Define

$$\ell(x) = \tau\mathbb{E}_{m,\epsilon}[\|\epsilon - \epsilon_{\hat{\theta}}(\sqrt{\bar{\alpha}_m}x + \sqrt{1-\bar{\alpha}_m}\epsilon, m)\|^2]]$$

Then, we have that $\hat{\mathcal{L}}(\hat{\theta}) = \frac{1}{n}\sum_{i=1}^{n}\ell(\mathbf{x}_0^{(i)})$ and $\mathcal{L}(\hat{\theta}) = \mathbb{E}_{\mathbf{x}_0}[\ell(\mathbf{x}_0)]$. Here, $\ell(\mathbf{x}_0^{(1)}), \ldots, \ell(\mathbf{x}_0^{(n)})$ are not independent since $\hat{\theta}$ is trained with the trajectories data $(\mathbf{x}_0^{(i)})_{i=1}^{n}$, which induces the dependence among $\ell(\mathbf{x}_0^{(1)}), \ldots, \ell(\mathbf{x}_0^{(n)})$. To deal with this dependence, we recall that

$$\mathbf{x}_0 = \begin{bmatrix} \mathbf{g}_0 & \mathbf{g}_1 & \cdots & \mathbf{g}_H \\ \mathbf{a}_0 & \mathbf{a}_K & \cdots & \mathbf{a}_{HK} \\ \mathbf{a}_1 & \mathbf{a}_{K+1} & \cdots & \mathbf{a}_{HK+1} \\ \vdots & \vdots & \ddots & \vdots \\ \mathbf{a}_{K-1} & \mathbf{a}_{2K-1} & \cdots & \mathbf{a}_{(H+1)K-1} \end{bmatrix} \in \mathcal{X} \subseteq \mathbb{R}^d,$$

where the baseline method is recovered by setting $K = 1$ (and hence $H = T/K = T$). To utilize this structure, we define $\mathcal{C}_k$ by

$$\mathcal{C}_k = \left\{ \mathbf{x}_0 = \begin{bmatrix} \mathbf{g}_0 & \mathbf{g}_1 & \cdots & \mathbf{g}_H \\ \mathbf{a}_0 & \mathbf{a}_K & \cdots & \mathbf{a}_{HK} \\ \mathbf{a}_1 & \mathbf{a}_{K+1} & \cdots & \mathbf{a}_{HK+1} \\ \vdots & \vdots & \ddots & \vdots \\ \mathbf{a}_{K-1} & \mathbf{a}_{2K-1} & \cdots & \mathbf{a}_{(H+1)K-1} \end{bmatrix} \in \mathcal{X} : k = \arg\max_{t \in [H]} \varphi(\mathbf{g}_t), \right\}.$$

We first write the expected error as the sum of the conditional expected error:

$$\mathbb{E}_{\mathbf{x}_0}[\ell(\mathbf{x}_0)] = \sum_k \mathbb{E}_{\mathbf{x}_0}[\ell(\mathbf{x}_0)|\mathbf{x}_0 \in \mathcal{C}_k]\Pr(\mathbf{x}_0 \in \mathcal{C}_k).$$

Similarly,

$$\frac{1}{n}\sum_{i=1}^{n}\ell(\mathbf{x}_0^{(i)}) = \frac{1}{n}\sum_{k \in I_\mathcal{K}}\sum_{i \in \mathcal{I}_k}\ell(\mathbf{x}_0^{(i)}) = \sum_{k \in I_\mathcal{K}}\frac{|\mathcal{I}_k|}{n}\frac{1}{|\mathcal{I}_k|}\sum_{i \in \mathcal{I}_k}\ell(\mathbf{x}_0^{(i)}),$$

where $\mathcal{I}_k = \{i \in [n] : \mathbf{x}_0^{(i)} \in \mathcal{C}_k\}$ and $I_\mathcal{K} = \{k \in [H] : |\mathcal{I}_k| \geq 1\}$. Using these, we decompose the difference into two terms:

$$\mathbb{E}_{\mathbf{x}_0}[\ell(\mathbf{x}_0)] - \frac{1}{n}\sum_{i=1}^{n}\ell(\mathbf{x}_0^{(i)}) = \sum_k \mathbb{E}_{\mathbf{x}_0}[\ell(\mathbf{x}_0)|\mathbf{x}_0 \in \mathcal{C}_k]\left(\Pr(\mathbf{x}_0 \in \mathcal{C}_k) - \frac{|\mathcal{I}_k|}{n}\right) \qquad (14)$$

$$+ \left(\sum_k \mathbb{E}_{\mathbf{x}_0}[\ell(\mathbf{x}_0)|\mathbf{x}_0 \in \mathcal{C}_k]\frac{|\mathcal{I}_k|}{n} - \frac{1}{n}\sum_{i=1}^{n}\ell(\mathbf{x}_0^{(i)})\right).$$

$$= \sum_k \mathbb{E}_{\mathbf{x}_0}[\ell(\mathbf{x}_0)|\mathbf{x}_0 \in \mathcal{C}_k]\left(\Pr(\mathbf{x}_0 \in \mathcal{C}_k) - \frac{|\mathcal{I}_k|}{n}\right)$$

$$+ \frac{1}{n}\sum_{k \in I_\mathcal{K}}|\mathcal{I}_k|\left(\mathbb{E}_{\mathbf{x}_0}[\ell(\mathbf{x}_0)|\mathbf{x}_0 \in \mathcal{C}_k] - \frac{1}{|\mathcal{I}_k|}\sum_{i \in \mathcal{I}_k}\ell(\mathbf{x}_0^{(i)})\right).$$

By following the proof of Lemma 5 of (Kawaguchi et al., 2023) and invoking Lemma 1 of (Kawaguchi et al., 2022), we have that for any $\delta > 0$, with probability at least $1 - \delta$,

$$\sum_k \mathbb{E}_{\mathbf{x}_0}[\ell(\mathbf{x}_0)|\mathbf{x}_0 \in \mathcal{C}_k]\left(\Pr(\mathbf{x}_0 \in \mathcal{C}_k) - \frac{|\mathcal{I}_k|}{n}\right) \qquad (15)$$

$$\leq \left(\sum_k \mathbb{E}_{\mathbf{x}_0}[\ell(\mathbf{x}_0)|\mathbf{x}_0 \in \mathcal{C}_k]\sqrt{\Pr(\mathbf{x}_0 \in \mathcal{C}_k)}\right)\sqrt{\frac{2\ln(H/\delta)}{n}}$$

$$\leq C\left(\sum_k \sqrt{\Pr(\mathbf{x}_0 \in \mathcal{C}_k)}\right)\sqrt{\frac{2\ln(H/\delta)}{n}}.$$

Here, note that for any $(f, h, M)$ such that $M > 0$ and $B \geq 0$ for all $X$, we have that $\mathbb{P}(f(X) \geq M) \geq \mathbb{P}(f(X) > M) \geq \mathbb{P}(Bf(X) + h(X) > BM + h(X))$, where the probability is with respect

to the randomness of $X$. Thus, by combining equation 14 and equation 15, we have that for any $\delta > 0$, with probability at least $1 - \delta$, the following holds:

$$\mathbb{E}_{\mathbf{x}_0}[\ell(\mathbf{x}_0)] - \frac{1}{n}\sum_{i=1}^{n}\ell(\mathbf{x}_0^{(i)}) \leq \frac{1}{n}\sum_{k\in I_{\mathcal{K}}}|\mathcal{I}_k|\left(\mathbb{E}_{\mathbf{x}_0}[\ell(\mathbf{x}_0)|\mathbf{x}_0\in\mathcal{C}_k] - \frac{1}{|\mathcal{I}_k|}\sum_{i\in\mathcal{I}_k}\ell(\mathbf{x}_0^{(i)})\right) \quad (16)$$

$$+ C\left(\sum_k\sqrt{\Pr(\mathbf{x}_0\in\mathcal{C}_k)}\right)\sqrt{\frac{2\ln(H/\delta)}{n}}$$

We now bound the first term in the right-hand side of equation equation 16. Define

$$\mathcal{H} = \{x \mapsto \tau\mathbb{E}_{m,\epsilon}[\|\epsilon - \epsilon_\theta(\sqrt{\bar{\alpha}_m}x + \sqrt{1-\bar{\alpha}_m}\epsilon, m)\|^2] : \theta\in\Theta\},$$

and

$$\mathcal{R}_t(\mathcal{H}) = \mathbb{E}_{(\mathbf{x}_0^{(i)})_{i=1}^n}\mathbb{E}_\xi\left[\sup_{h\in\mathcal{H}}\frac{1}{|\mathcal{I}_t|}\sum_{i=1}^{|\mathcal{I}_t|}\xi_i h(\mathbf{x}_0^{(i)}) \mid \mathbf{x}_0^{(i)}\in\mathcal{C}_t\right].$$

with independent uniform random variables $\xi_1,\ldots,\xi_n$ taking values in $\{-1,1\}$. We invoke Lemma 4 of (Pham et al., 2021) to obtain that for any $\delta > 0$, with probability at least $1 - \delta$,

$$\frac{1}{n}\sum_{k\in I_{\mathcal{K}}}|\mathcal{I}_k|\left(\mathbb{E}_{\mathbf{x}_0}[\ell(\mathbf{x}_0)|\mathbf{x}_0\in\mathcal{C}_k] - \frac{1}{|\mathcal{I}_k|}\sum_{i\in\mathcal{I}_k}\ell(\mathbf{x}_0^{(i)})\right) \quad (17)$$

$$\leq \frac{1}{n}\sum_{k\in I_{\mathcal{K}}}|\mathcal{I}_k|\left(2\mathcal{R}_k(\mathcal{H}) + C\sqrt{\frac{\ln(H/\delta)}{2|\mathcal{I}_k|}}\right)$$

$$= \sum_{k\in I_{\mathcal{K}}}\frac{2|\mathcal{I}_k|\mathcal{R}_k(\mathcal{H})}{n} + C\sqrt{\frac{\ln(H/\delta)}{2n}}\sum_{k\in I_{\mathcal{K}}}\sqrt{\frac{|\mathcal{I}_k|}{n}}$$

$$\leq \sum_{k\in I_{\mathcal{K}}}\frac{2|\mathcal{I}_k|\mathcal{R}_k(\mathcal{H})}{n} + C\sqrt{\frac{H\ln(H/\delta)}{2n}},$$

where the last line follows from the Cauchy–Schwarz inequality applied on the term $\sum_{k\in I_{\mathcal{K}}}\sqrt{\frac{|\mathcal{I}_k|}{n}}$ as

$$\sum_{k\in I_{\mathcal{K}}}\sqrt{\frac{|\mathcal{I}_k|}{n}} \leq \sqrt{\sum_{k\in I_{\mathcal{K}}}\frac{|\mathcal{I}_k|}{n}}\sqrt{\sum_{k\in I_{\mathcal{K}}}1} = \sqrt{\sum_{k\in I_{\mathcal{K}}}1} \leq \sqrt{H}.$$

On the other hand, by using Jensen's inequality,

$$\frac{1}{H}\sum_{k=1}^{H}\sqrt{\Pr(\mathbf{x}_0\in\mathcal{C}_k)} \leq \sqrt{\frac{1}{H}\sum_{k=1}^{H}\Pr(\mathbf{x}_0\in\mathcal{C}_k)} = \frac{1}{\sqrt{H}}$$

which implies that

$$\sum_{k=1}^{H}\sqrt{\Pr(\mathbf{x}_0\in\mathcal{C}_k)} \leq \sqrt{H}. \quad (18)$$

By combining equations equation 16 and equation 17 with union bound along with equation 18, it holds that any $\delta > 0$, with probability at least $1 - \delta$,

$$\mathbb{E}_{\mathbf{x}_0}[\ell(\mathbf{x}_0)] - \frac{1}{n}\sum_{i=1}^{n}\ell(\mathbf{x}_0^{(i)})$$

$$\leq \sum_{k\in I_{\mathcal{K}}}\frac{2|\mathcal{I}_k|\mathcal{R}_k(\mathcal{H})}{n} + C\sqrt{\frac{H\ln(2H/\delta)}{2n}} + C\left(\sum_k\sqrt{\Pr(\mathbf{x}_0\in\mathcal{C}_k)}\right)\sqrt{\frac{2\ln(2H/\delta)}{n}}$$

$$\leq \sum_{k\in I_{\mathcal{K}}}\frac{2|\mathcal{I}_k|\mathcal{R}_k(\mathcal{H})}{n} + C\left(\sqrt{2}^{-1} + \sqrt{2}\right)\sqrt{\frac{H\ln(2H/\delta)}{n}}$$

Since $H \leq \lceil T/K \rceil$, this implies that any $\delta > 0$, with probability at least $1 - \delta$,

$$\mathbb{E}_{\mathbf{x}_0}[\ell(\mathbf{x}_0)] - \frac{1}{n}\sum_{i=1}^{n}\ell(\mathbf{x}_0^{(i)}) \leq C_0\sqrt{\left\lceil \frac{T}{K} \right\rceil \frac{\ln(\lceil \frac{T}{K} \rceil \frac{2}{\delta})}{n}} + \sum_{t \in I_\mathcal{K}} \frac{2|\mathcal{I}_t|\mathcal{R}_t(\mathcal{H})}{n}.$$

where $C_0 = C\left(\sqrt{2}^{-1} + \sqrt{2}\right)$. This proves the first statement of this theorem. $\qquad\square$

## J.2 Proof of Proposition 1

*Proof.* For the second statement, let $K = 1$ and we consider the effect of increasing $K$ from one to an arbitrary value greater than one. Denote by $\mathcal{R}_t(\mathcal{H})$ and $\tilde{\mathcal{R}}_t(\tilde{\mathcal{H}})$ the conditional Rademacher complexities for $K = 1$ (base case) and $K > 1$ (after increasing $K$) respectively: i.e., we want to show that $\mathcal{R}_t(\mathcal{H}) \geq \tilde{\mathcal{R}}_t(\tilde{\mathcal{H}})$. Given the increasing value of $K$, let $t \in \{1, \dots, T\}$ such that $s_t$ is not skipped after increasing $K$. From the definition of $\mathcal{H}$,

$$\mathcal{R}_t(\mathcal{H}) = \mathbb{E}_{(\mathbf{x}_0^{(i)})_{i=1}^n} \mathbb{E}_\xi \left[ \sup_{h \in \mathcal{H}} \frac{1}{|\mathcal{I}_t|} \sum_{i=1}^{|\mathcal{I}_t|} \xi_i h(\mathbf{x}_0^{(i)}) \mid \mathbf{x}_0^{(i)} \in \mathcal{C}_t \right] \tag{19}$$

$$= \mathbb{E}_{(\mathbf{x}_0^{(i)})_{i=1}^n} \mathbb{E}_\xi \left[ \sup_{\theta \in \Theta} \frac{1}{|\mathcal{I}_t|} \sum_{i=1}^{|\mathcal{I}_t|} \xi_i \mathbb{E}_{m,\epsilon}[\|\epsilon - \epsilon_\theta(\varsigma(\mathbf{x}_0^{(i)}), m)\|^2] \mid \mathbf{x}_0^{(i)} \in \mathcal{C}_t \right].$$

where everything is defined for $K = 1$ and $\varsigma(\mathbf{x}_0^{(i)}) = \sqrt{\bar{\alpha}_m}\mathbf{x}_0^{(i)} + \sqrt{1 - \bar{\alpha}_m}\epsilon$. Here, we recall that $\epsilon_\theta(\mathbf{x}_m, m) = Wg(\mathbf{x}_m, m)$ for some function $g$ and an output layer weight matrix $W$ such that the parameters of $g$ does not contain the entries of the output layer weight matrix $W$. This implies that $\epsilon_\theta(\varsigma(\mathbf{x}_0^{(i)}), m) = W\tilde{g}_m(V\mathbf{x}_0^{(i)})$ where $\tilde{g}_m(x) = g(\tilde{\varsigma}(x), m)$ where $\tilde{\varsigma}(x) = \sqrt{\bar{\alpha}_m}x + \sqrt{1 - \bar{\alpha}_m}V\epsilon$, and that we can decompose $\Theta = \mathcal{W} \times \mathcal{V} \times \tilde{\Theta}$ with which $\theta$ can be decomposed into $W \in \mathcal{W}$, $V \in \mathcal{V}$, and $\tilde{\theta} \in \tilde{\Theta}$. Using this,

$$\mathcal{R}_t(\mathcal{H}) = \mathbb{E}_{(\mathbf{x}_0^{(i)})_{i=1}^n} \mathbb{E}_\xi \left[ \sup_{\theta \in \Theta} \frac{1}{|\mathcal{I}_t|} \sum_{i=1}^{|\mathcal{I}_t|} \xi_i \mathbb{E}_{m,\epsilon}[\|\epsilon - W\tilde{g}_m(V\mathbf{x}_0^{(i)})\|^2] \mid \mathbf{x}_0^{(i)} \in \mathcal{C}_t \right] \tag{20}$$

$$= \mathbb{E}_{(\mathbf{x}_0^{(i)})_{i=1}^n} \mathbb{E}_\xi \left[ \sup_{(W,V,\tilde{\theta}) \in \mathcal{W} \times \mathcal{V} \times \tilde{\Theta}} \frac{1}{|\mathcal{I}_t|} \sum_{i=1}^{|\mathcal{I}_t|} \xi_i \sum_{j=1}^{d} \mathbb{E}_{m,\epsilon}[(\epsilon_j - W_j\tilde{g}_m(V\mathbf{x}_0^{(i)}))^2] \mid \mathbf{x}_0^{(i)} \in \mathcal{C}_t \right].$$

where $W_j$ is the $j$-th row of $W$. Recall that when we increase $K$, some states are skipped and accordingly $d$ decreases. Let $d_0$ be the $d$ after $K$ increased from one to some value greater than one: i.e., $d_0 \leq d$. Without loss of generality, let us arrange the order of the coordinates over $j = 1, 2 \dots, d_0, d_0 + 1, \dots, d$ so that $j = d_0 + 1, d_0 + 2, \dots, d$ are removed after $K$ increases.

Since $\Theta$ contains $\theta$ with $W$ and $V$ such that $\|W\|_\infty \leq \zeta_W$ and $\|V\|_\infty < \zeta_V$ for some $\zeta_W$ and $\zeta_V$, the set $\mathcal{W}$ contains $W$ such that $W_j = 0$ for $j = d_0 + 1, d_0 + 2, \dots, d$. Define $\mathcal{W}_0$ such that $\mathcal{W} = \mathcal{W}_0 \times \tilde{\mathcal{W}}_0$ where $(W_j)_{j=1}^{d_0} \in \mathcal{W}_0$ and $(W_j)_{j=d_0+1}^{d} \in \tilde{\mathcal{W}}_0$. Notice that $\mathcal{W} = \{(W_j)_{j=1}^{d} : \|(W_j)_{j=1}^{d}\|_\infty \leq \zeta_W\}$ and $\mathcal{W}_0 = \{(W_j)_{j=1}^{d_0} : \|(W_j)_{j=1}^{d_0}\|_\infty \leq \zeta_W\}$. Since we take supremum over $W \in \mathcal{W}$, setting $W_j = 0$ for $j = d_0 + 1, d_0 + 2, \dots, d$ attains a lower bound as

$$\mathcal{R}_t(\mathcal{H}) = \mathbb{E}_{(\mathbf{x}_0^{(i)})_{i=1}^n} \mathbb{E}_\xi \left[ \sup_{(W,V,\tilde{\theta}) \in \mathcal{W} \times \mathcal{V} \times \tilde{\Theta}} \frac{1}{|\mathcal{I}_t|} \sum_{i=1}^{|\mathcal{I}_t|} \xi_i \sum_{j=1}^{d} \mathbb{E}_{m,\epsilon}[(\epsilon_j - W_j\tilde{g}_m(V\mathbf{x}_0^{(i)}))^2] \mid \mathbf{x}_0^{(i)} \in \mathcal{C}_t \right]$$

$$\geq \mathbb{E}_{(\mathbf{x}_0^{(i)})_{i=1}^n} \mathbb{E}_\xi \left[ \sup_{(W,V,\tilde{\theta}) \in \mathcal{W}_0 \times \mathcal{V} \times \tilde{\Theta}} \frac{1}{|\mathcal{I}_t|} \sum_{i=1}^{|\mathcal{I}_t|} \xi_i A_i \mid \mathbf{x}_0^{(i)} \in \mathcal{C}_t \right]$$

$$= \mathbb{E}_{(\mathbf{x}_0^{(i)})_{i=1}^n} \mathbb{E}_\xi \left[ \sup_{(W,V,\tilde{\theta}) \in \mathcal{W}_0 \times \mathcal{V} \times \tilde{\Theta}} \frac{1}{|\mathcal{I}_t|} \sum_{i=1}^{|\mathcal{I}_t|} \xi_i \sum_{j=1}^{d_0} \mathbb{E}_{m,\epsilon}[(\epsilon_j - W_j\tilde{g}_m(V\mathbf{x}_0^{(i)}))^2] \mid \mathbf{x}_0^{(i)} \in \mathcal{C}_t \right]$$

where $A_i = \sum_{j=1}^{d_0} \mathbb{E}_{m,\epsilon}[(\epsilon_j - W_j \tilde{g}_m(V \mathbf{x}_0^{(i)}))^2] + \sum_{j=d_0+1}^{d} \mathbb{E}_{m,\epsilon}[(\epsilon_j)^2]$ and the last line follows from the fact that

$$\mathbb{E}_\xi \sup_{(W,\tilde{\theta}) \in \mathcal{W} \times \tilde{\Theta}} \sum_{j=d_0+1}^{d} \xi_i \mathbb{E}_{m,\epsilon}[(\epsilon_j)^2] = \mathbb{E}_\xi \sum_{j=d_0+1}^{d} \xi_i \mathbb{E}_{m,\epsilon}[(\epsilon_j)^2] = \sum_{j=d_0+1}^{d} \mathbb{E}_\xi[\xi_i] \mathbb{E}_{m,\epsilon}[(\epsilon_j)^2] = 0.$$

Similarly, since $\Theta$ contains $\theta$ with $W$ and $V$ such that $\|W\|_\infty \leq \zeta_W$ and $\|V\|_\infty < \zeta_V$ for some $\zeta_W$ and $\zeta_V$, the set $\mathcal{V}$ contains $V$ such that $V_j = 0$ for $j = d_0 + 1, d_0 + 2, \ldots, d$, where $V_j$ is the $j$-th row of $V$. Define $\mathcal{V}_0$ such that $\mathcal{V} = \mathcal{V}_0 \times \tilde{\mathcal{V}}_0$ where $(V_j)_{j=1}^{d_0} \in \mathcal{V}_0$ and $(V_j)_{j=d_0+1}^{d} \in \tilde{\mathcal{V}}_0$ . Notice that $\mathcal{V} = \{(V_j)_{j=1}^{d} : \|(V_j)_{j=1}^{d}\|_\infty \leq \zeta_V\}$ and $\mathcal{V}_0 = \{(V_j)_{j=1}^{d_0} : \|(V_j)_{j=1}^{d_0}\|_\infty \leq \zeta_V\}$. Since we take supremum over $V \in \mathcal{V}$, setting $V_j = 0$ for $j = d_0 + 1, d_0 + 2, \ldots, d$ attains a lower bound as

$$\mathcal{R}_t(\mathcal{H}) \geq \mathbb{E}_{(\mathbf{x}_0^{(i)})_{i=1}^n} \mathbb{E}_\xi \left[ \sup_{(W,V,\tilde{\theta}) \in \mathcal{W}_0 \times \mathcal{V} \times \tilde{\Theta}} \frac{1}{|\mathcal{I}_t|} \sum_{i=1}^{|\mathcal{I}_t|} \xi_i \sum_{j=1}^{d_0} \mathbb{E}_{m,\epsilon}[(\epsilon_j - W_j \tilde{g}_m(V \mathbf{x}_0^{(i)}))^2] \mid \mathbf{x}_0^{(i)} \in \mathcal{C}_t \right]$$

$$= \mathbb{E}_{(\mathbf{x}_0^{(i)})_{i=1}^n} \mathbb{E}_\xi \left[ \sup_{(W,V,\tilde{\theta}) \in \mathcal{W}_0 \times \mathcal{V} \times \tilde{\Theta}} \frac{1}{|\mathcal{I}_t|} \sum_{i=1}^{|\mathcal{I}_t|} \xi_i B_i(d) \mid \mathbf{x}_0^{(i)} \in \mathcal{C}_t \right]$$

$$\geq \mathbb{E}_{(\mathbf{x}_0^{(i)})_{i=1}^n} \mathbb{E}_\xi \left[ \sup_{(W,V,\tilde{\theta}) \in \mathcal{W}_0 \times \mathcal{V}_0 \times \tilde{\Theta}} \frac{1}{|\mathcal{I}_t|} \sum_{i=1}^{|\mathcal{I}_t|} \xi_i B_i(d_0) \mid \mathbf{x}_0^{(i)} \in \mathcal{C}_t \right]$$

$$= \mathbb{E}_{(\tilde{x}_0^{(i)})_{i=1}^n} \mathbb{E}_\xi \left[ \sup_{(\tilde{W},\tilde{V},\tilde{\theta}) \in \mathcal{W}_0 \times \mathcal{V}_0 \times \tilde{\Theta}} \frac{1}{|\mathcal{I}_t|} \sum_{i=1}^{|\mathcal{I}_t|} \xi_i \mathbb{E}_{m,\epsilon}[\|\tilde{\epsilon} - \tilde{W} \tilde{g}_m(\tilde{V} \tilde{x}_0^{(i)})\|^2] \mid \tilde{x}_0^{(i)} \in \tilde{\mathcal{C}}_t \right]$$

$$\geq \tilde{\mathcal{R}}_t(\tilde{\mathcal{H}})$$

where $B_i(d) = \sum_{j=1}^{d_0} \mathbb{E}_{m,\epsilon} \left[ \left( \epsilon_j - W_j \tilde{g}_m \left( \sum_{k=1}^{d} V_k (\mathbf{x}_0^{(i)})_k \right) \right)^2 \right]$, $\tilde{\epsilon} = (\epsilon_j)_{j=1}^{d_0}$, $\tilde{x}_0^{(i)} = ((\tilde{x}_0^{(i)})_j)_{j=1}^{d_0}$, $\tilde{\mathcal{C}}_t$ is the $\mathcal{C}_t$ for $\tilde{x}_0^{(i)}$ with skipping states, and $\tilde{\mathcal{R}}_t(\tilde{\mathcal{H}})$ is the conditional Rademacher complexity after increasing $K > 1$. The last line follows from the same steps of equation 19 and equation 20 applied for $\tilde{\mathcal{R}}_t(\tilde{\mathcal{H}})$ and the fact that $|\mathcal{I}_t|$ of $\mathcal{R}_t(\mathcal{H})$ is smaller than that of $\tilde{\mathcal{R}}_t(\tilde{\mathcal{H}})$ (due to the effect of removing the states), along with the assumption that $\mathcal{R}_t(\mathcal{H})$ does not increase when we increase $n_t$. This proves the second statement.

$\square$

