# OpenReview forum: "Simple Hierarchical Planning with Diffusion"
_ICLR.cc/2024/Conference — ICLR 2024 poster_

### Official Review · Reviewer_KFqH · 2023-10-30

**Soundness:** 3 good
**Presentation:** 3 good
**Contribution:** 3 good
**Rating:** 6
**Confidence:** 3

**Summary:**

The paper proposes an hierarchical variant of Diffuser. The algorithm involves using a high-level planner that plans trajectories in a sparser manner and a low-level planner that adds detailed states to the trajectories produces by the high-level planner. Experiment result demonstrate that hierarchical planning produces better performance in long-horizon planning tasks and offline RL tasks. The authers also study the impact of kernel size in Diffuser and the impact of the jumpy steps. An interesting observation is that Diffuser with a small receptive field can not cover the data distribution well.

**Strengths:**

1. The paper is well-written and easy to follow.
2. The method is straightforward and demonstrates strong results in long-horizon planning tasks, especially the AntMaze.
3. The study of relationship between receptive field of the diffusion model and the data coverage ratio is interesting and offers valuable insight for futher research on diffusion models for planning in more complex domains.

**Weaknesses:**

One minor weakness is that the baseline Decision Diffuser[1] is missing. Also the study presented in this paper focuses on using the CNN architecture as Diffuser does. A comparison between using CNN and transformer architecutre, such as Decision Diffuser, would be encouraged.

[1] Ajay, Anurag, et al. "Is conditional generative modeling all you need for decision-making?." ICLR 2023.

**Questions:**

1. According to Fig 3., Diffuser with a small kernel size covers one mode of the data distribution during actual execution. Is it the case that the standard Diffuser is able to \textbf{cover} the full data distribution but just fails to achieve full data coverage during actual execution? If so, what might be the cause the such gap?
2. Still concerning Fig 3., although HD seems to cover most of the data distribution, HD fails to cover two short segments/branches. Why does such pheomena occur?

---

> ### Author Response · Authors · 2023-11-23
> **Response to Reviewer**
>
> ## R1: Baseline Decision Diffuser (DD) is missing
> Thank you for pointing out the Decision Diffuser (DD) omission as a baseline in our comparative analysis. We present the results in Table 1 below and will include DD in Table 2 in our revised paper.
>
> Table 1. Comparison between DD and HD
> | Task | HD | DD |
> | --- | :-: | :-: |
> | MedExp-HalfCheetah | 92.5 +- 0.3 | 90.6 +- 1.3 |
> | MedExp-Hopper | 115.3 +- 1.1 | 111.8 +- 1.8 |
> | MedExp-Walker2d | 107.1 +/- 0.1 | 108.8 +- 1.7 |
> | Medium-HalfCheetah | 46.7 +- 0.2 | 49.1 +- 1.0 |
> | Medium-Hopper | 99.3 +- 0.3 | 79.3 +- 3.6 |
> | Medium-Walker2d | 84.0 +- 0.6 | 82.5 +- 1.4 |
> | MedRep-HalfCheetah | 38.1 +- 0.7 | 39.3 +- 4.1 |
> | MedRep-Hopper | 94.7 +- 0.7 | 100.0 +- 0.7 |
> | MedRep-Walker2d | 84.1 +- 2.2 | 75.0 +- 4.3 |
> | Gym Average | **84.6** | 81.8 |
>
> ## R2: A comparison between using CNN and transformer architecture, such as Decision Diffuser, would be encouraged.
> We appreciate the reviewer's constructive suggestion. As suggested, we compared our model (based on U-Net (CNN)) with Transformer, and the results are in Table 2 below.
>
> Table 2: Comparison on HD-UNet and HD-Transformer
> | Task | HD-UNet | HD-Transformer  |
> | --- | :-: | :-: |
> | Maze2d-Large | 128.4 +/- 3.63 | 127.9 +- 3.2 |
> | Maze2d-Medium | 135.6 +/- 3.00 | 136.1 +- 2.6 |
> | Maze2d-UMaze | 155.8 +/- 2.54 | 154.1 +- 3.6 |
> | Maze2d Average | **139.9** | 139.4 |
> | MedExp-HalfCheetah | 92.5 +- 0.3 | 88.4 +- 0.6 |
> | MedExp-Hopper | 115.3 +- 1.1 | 103.9 +- 5.9 |
> | MedExp-Walker2d | 107.1 +/- 0.1 | 107.0 +- 0.3 |
> | Medium-HalfCheetah | 46.7 +- 0.2 | 45.3 +- 0.5 |
> | Medium-Hopper | 99.3 +- 0.3 | 94.0 +- 5.4 |
> | Medium-Walker2d | 84.0 +- 0.6 | 82.8 +- 1.7 |
> | MedRep-HalfCheetah | 38.1 +- 0.7 | 39.5 +- 0.2 |
> | MedRep-Hopper | 94.7 +- 0.7 | 91.4 +- 1.5 |
> | MedRep-Walker2d | 84.1 +- 2.2 | 81.2 +- 1.1 |
> | Gym Average | **84.6** | 81.5 |
>
> For this experiment, we use the hyperparameter setting in the Decision Transformer (DT) as a starting point for our investigation. The results, as shown in the table, reveal that the HD-Transformer achieves similar performance to the HD-UNet in Maze2D tasks, though it is slightly less effective in the Gym-MuJoCo tasks. While the HD-Transformer shows promise, we would like to emphasize that our primary contribution is not the backbone architecture but the benefits of hierarchical structures. Nonetheless, we are grateful for the reviewer's insightful and constructive recommendation. This ablation study will make the paper better.
>
> Regarding the backbone architecture, we also did another related experiment suggested by another reviewer. Here, the goal is to see the performance of a deeper multi-scale U-Net as it can capture the hierarchical structure better within one architecture. As shown in Table 3 below, we see that HD still works significantly better. For more details, please see response R3 to reviewer zGcK.
>
> Table 3: Comparison on Multiscale U-Net and HD
> | Task | Multiscale U-Net | HD |
> | --- | :-: | :-: |
> | Umaze | 114.0 +- 4.6 | 128.4±3.6 |
> | Medium | 117.8 +- 1.9 | 135.6±3.0 |
> | Large | 124.8 +- 6.0 | 155.8±2.5 |
>
> ## Q1: According to Fig 3., Diffuser with a small kernel size covers one mode of the data distribution during actual execution. Is it the case that the standard Diffuser is able to cover the full data distribution but just fails to achieve full data coverage during actual execution? If so, what might be the cause the such gap?
>
> We appreciate the reviewer's query, which we also find interesting. The answer is no. We found that the diffuser generates plans in the bottom half and actual execution following this plan.
>
> We think the chance that the actions deviate from the generated plan is low. This is primarily because the Diffuser model employs a PD controller to derive actions from the planned states, which include position and velocity. Consequently, the actual rollout is expected to follow the generated plan closely.
>
> ## Q2: In Fig 3, HD fails to cover two short segments/branches. Why does such phenomena occur?
> Thank you for highlighting this observation. It is a very interesting phenomenon to us as well. Our current best hypothesis is that it has something to do with ignoring actions in the Diffusion modeling. First, in another related work, AdaptDiffuser [1], which also only models states, we observe a similar smoothing effect in Figure 3. (a) and (b). Second, we observed that when we add actions in the diffuser (HD-DA), the smoothing effect is mitigated, as shown in the figure [here](https://anonymous.4open.science/api/repo/plan_with_HD-E821/file/data_coverage_HDDA.png). These suggest that the smoothing is highly related to ignoring actions in diffusion modeling. Thank you for pointing out this interesting question. We will discuss this in the revised paper.

---

### Official Review · Reviewer_AgLN · 2023-10-31

**Soundness:** 3 good
**Presentation:** 3 good
**Contribution:** 2 fair
**Rating:** 6
**Confidence:** 4

**Summary:**

The authors of this paper present Hierarchical Diffuser, a simple yet effective hierarchical planning approach using diffusion models to improve the performance and efficiency of diffusion-based planning for long-horizon tasks. By utilizing a two-pronged structure, the Hierarchical Diffuser combines a high-level planner, the Sparse Diffuser, to generate sub-goals, and a low-level planner to refine the plan. This approach significantly outperforms non-hierarchical planning methods, while also enhancing computational efficiency in training and planning speed. Furthermore, the method displays improved generalization capabilities on out-of-distribution tasks compared to non-hierarchical Diffusers.

**Strengths:**

- Originality: This work creatively combines the strengths of hierarchical planning and diffusion models, achieving significant improvements in performance and computational efficiency compared to existing methods.
- Quality: The contribution provides a thorough theoretical analysis of the method, backed by well-designed experiments and in-depth analyses. The authors emphasize the rationale behind the design decisions and the resultant benefits of their approach.
- Clarity: The paper is well-organized, clearly written, and explains the method along with its advantages and limitations in a coherent and accessible way.

**Weaknesses:**

- Considering the existence of the HDMI algorithm, the contribution of this paper are limited. However, compared to HDMI, this paper has conducted extensive experiments on generalization and provided corresponding theoretical support. One suggestion I have is that the authors can emphasize the advantages of Hierarchical Diffuser from the perspective of generalization, and highlight the algorithm's generalization performance in the experimental design.
- In the hierarchical framework, the quality of the generated high-level goals is crucial to the effectiveness of the algorithm. The authors emphasize that the Hierarchical Diffuser is simpler compared to existing hierarchical methods but have only designed one goal selection method. In addition to the fixed time interval heuristic method used in this paper, HDMI also mentions two heuristic methods based on spatial intervals and reward scale, which are equally simple and have low implementation costs. If the authors could take these heuristic methods into consideration, I believe it would have a positive impact on the completeness of the paper's content. Furthermore, the demonstration of the quality of the generated goals is essential, and there is a gap in this aspect in the current paper.

**Questions:**

- I have noticed that both Diffuser and Decision Diffuser employ “inpainting” techniques for trajectory generation in goal-conditioned tasks and the lower-level planner of HDMI. However, this paper utilizes return-guided sampling. What were the considerations behind the authors' choice between the two methods?
- This paper proposes to get the sub-goals first with the planning-based subgoal extractor and then train HDMI with the sub-goals as supervision. Although this two-step process makes sense, the overall training process becomes ad hoc and less general. Is there any connection between the upper and lower layers of Hierarchical Diffuser's planner? Can it be trained end-to-end?

---

> ### Author Response · Authors · 2023-11-22
> **Response to Reviewer**
>
> ## R1: Highlight the algorithm’s generalization performance more
>
> We thank the insightful feedback. Following the suggestion, we conducted an additional experiment for the OOD generalization task. Please see response R4 for reviewer zGcK for the details.
>
> ## R2 Considering HDMI, contribution is limited in this work.
>
> Thank you for your feedback regarding the perceived limited contribution of our work in comparison to HDMI. While there are certain similarities between our HD framework and HDMI, the differences in focus, methodology, and resultant performance and efficiency gains highlight the substantial contributions of our work. Overall, our paper leads to a different lesson from HDMI because our paper suggests that **the high complexity of HDMI is actually not required**; **we can still achieve even better performance with a much simpler approach**.
>
> In the following ,we respectfully offer a different perspective, highlighting several distinct aspects of our work:
>
> - **Different methodologies:**
>     - **Problem formulation**: HDMI casts the offline decision-making as a conditional generative problems where sampling is conditioned on the pre-defined desired return. Our method formulate the decision-making problems as sequence modeling problem. Policy is improved via the classifier guidance sampling, which guides the planner to generate plans that maximize the expected return.
>     - **Sub-goal selection**: HDMI’s sub-goal preprocessing requires explicitly building a graph on the training dataset and applying costly graph search. Our method works robustly with simple fixed time interval heuristic method, which is more computationally efficient. HDMI’s subgoal selection is very complex. This process requires components like (1) training a separate RL policy like IQN or D4PG only for subgoal selection, (2) building a weighted directed graph, (3) dataset reduction by running K-means++ algorithm, (4) learning goal-conditioned value function, (5) distributional RL, (6) running Floyd-Warshall algorithm to find shortest path, (7) low-temperature sampling to prevent “same sub-gaol issue”, and so on. Also, code is not available. These are all used only for goal selection.
>     - **Low-level planning**: HDMI low-level planner is ignorant of the task rewards, which might lead to local sub-optimal. Our low-level planner is guided with both sub-goals and the task rewards.
>     - **Policy inference**: HDMI models only the states sequence, actions are inferred from an inverse dynamic module. Our method models both states and actions directly.
>     - **OOD generalization**: We presented one compositional generalization task in our paper, the solvability of which by HDMI remains unclear. During rebuttal, we provide an additional OOD collision avoidance task, a challenge that conditional-generative methods like HDMI can not address effectively, even with ground-truth reward function. This is because the OOD obstacles introduced novel reward structures that will shift the distribution of trajectory returns, i.e. a novel trajectory return value has never been seen during training, making return-conditioned generating fail. However, the classifier guidance used in our method can guide the plans gradually move away from collision.
>     - **Theoretic analysis:** We also provide theoretical analysis which fits well to the proposed model due to the simplicity of the proposed method.
> - **Superior Performance:** Our empirical results, as shown in Table 1 and Table 2, HD significantly out-performs HDMI on both the long-horizon goal-conditioned tasks and Gym-MuJoCo tasks.
>
> ## R3: Demonstration of the quality of the generated goals is missing, which is essential.
>
> We thank the reviewer for your valuable feedback highlighting the need to demonstrate the quality of the generated sub-goals. To address this, we have included visualizations of randomly sampled plans from the Maze2D tasks, which can be found [here](https://anonymous.4open.science/api/repo/plan_with_HD-E821/file/maze2d-subgoal.png). In these visualizations, the sub-goals are highlighted with large circles. These illustrations clearly demonstrate how the sub-goals effectively guide the low-level plan towards the overall goal. We appreciate the opportunity to enhance the clarity and depth of our presentation.

---

> ### Author Response · Authors · 2023-11-22
> **Continued - Response to Reviewer**
>
> ## R4: Goal-selection method is not flexible (only fixed time interval), may consider other methods such as spatial intervals and reward scale as in HDMI
> Thank you for the reviewer’s suggestion regarding the flexibility of our goal-selection method. We ran additional experiments on several alternatives of sub-goal selection.
>
> - Route Sampling[1] (RS): We also consider choosing waypoints with fixed length intervals as sub-goals in line with HDMI. Specifically, denote the distance moved after action $a_t$ as $\delta_t$. Then, the route length can be computed as $S = \sum_{t=0}^{T −1} \delta_t$ . We pick the waypoints with a fixed interval of $S/k$, where $k$ is the number of sub-goals.
> - Value Sampling[2] (VS): Also inspired by HDMI, we test the value sampling method, where the most valuable states are chosen as sub-goals. Specifically, the distance-weighted accumulated reward is used to value each state after state $s_i$: $W(s_j) = \sum_{k=i+1}^j \frac{r_k}{j-i}$.
> - Future Sampling[3] (FS): Beyond RS and VS, we also explored a hindsight heuristic method, randomly selecting future states as sub-goals.
>
> Notably, in RS and VS, certain states might never be chosen as sub-goals, unlike FS and the fixed time interval sampling (TS) used in HD, which offers equal probability for each state to be selected as a sub-goal.
>
> Given the varying lengths of sub-tasks generated by these selection methods, integrating dense action at the high level was impractical. Hence, we focused our experiments on HD rather than HD-DA. At the low level, sub-trajectories were padded to a consistent length $L$. It's important to note that excluding dense action data at the high level may hinder the learning of the value function, potentially leading to a marginal decrease in performance, as shown in Table 4 in the paper. As presented in the table below, the results demonstrate that our hierarchical framework is generally resilient across different sub-goal selection methods. While HD-VS and HD-RS exhibited somewhat lower performance, we hypothesize this may be due to uneven sampling of valuable states, which could impact the planning guidance function's effectiveness.
> |  Task | HD-DA | HD | HD-FS | HD-VS | HD-RS |
> | --- | :-: | :-: | :-: | :-: | :-: |
> | MedExp-Halfcheetah | 92.5 +/- 0.3 | 92.1 +- 0.5 | 87.6 +- 0.7 | 87.6 +- 0.6 | 88.4 +- 0.4 |
> | MedExp-Hopper | 115.3 +/- 1.1 | 104.1 +- 8.2 | 106.5 +- 5.5 | 108.9 +- 4.8 | 106.4 +- 5.0 |
> | MedExp-Walker2d | 107.1 +/- 0.1 | 107.4 +- 0.3 | 107.0 +- 0.1 | 107.4 +- 0.2 | 107.4 +- 0.3 |
> | Medium-Halfcheetah | 46.7 +/- 0.2 | 45.2 +- 0.2 | 43.9 +- 0.4 | 43.2 +- 0.3 | 43.6 +- 0.9 |
> | Medium-Hopper | 99.3 +/- 0.3 | 99.2 +- 0.7 | 100.9 +- 0.8 | 92.3 +- 4.2 | 95.8 +- 1.3 |
> | Medium-Walker2d | 84.0 +/- 0.6 | 82.6 +- 0.8 | 83.1 +- 1.0 | 82.4 +- 0.9 | 82.9 +- 1.1 |
> | MedRep-Halfcheetah | 38.1 +/- 0.7 | 37.5 +- 1.7 | 39.7 +- 0.3 | 38.1 +- 0.7 | 38.4 +- 0.8 |
> | MedRep-Hopper | 94.7 +/- 0.6 | 93.4 +- 3.1 | 90.9 +- 1.7 | 91.3 +/- 1.3 | 92.6 +- 1.2 |
> | MedRep-Walker2d | 84.1 +/- 2.2 | 77.2 +- 3.3 | 80.9 +- 1.7 | 75.7 +- 2.10 | 76.4 +/- 2.7 |
> | Average | **84.64** | **82.1** | **82.3** | 80.8 | 81.3 |
>
> [1]: Lai, Y., Wang, W., Yang, Y., Zhu, J., and Kuang, M. Hindsight planner.
>
> [2]: Correia, A. and Alexandre, L. A. Hierarchical decision transformer.
>
> [3]: Marcin Andrychowicz, et al. Hindsight Experience Replay
>
> ## Q1: Why only return-guided sampling is used in this paper instead of inpainting?
> As shown in Appendix B, we utilized both value-guided sampling and inpainting, aligning with the approach used in Diffuser. Specifically, for Maze tasks, we employed inpainting as our sampling strategy. In the case of more complex controlling tasks, we augmented this with value guidance.

---

> > ### Author Response · Authors · 2023-11-22
> > **Continued - Continued - Response to Reviewer**
> >
> > ## Q2: During training, sub-goal is first sampled from high-level planner, then train HDMI with sub-goals as supervision. Is there connections between the upper and lower layers of HD’s planner? Can it be trained end-to-end?
> >
> > Thank you for the questions. There is a bit of a misunderstanding to address first. In our model, both the high-level and low-level planners are trained simultaneously, not in a sequential manner. As shown in Figure 1, the high-level planner is designed to model the distribution of sub-goal sequences, whereas the low-level planner is responsible for modeling sub-trajectories, not sub-goals. Our model does not incorporate HDMI; instead, the high-level and low-level planners are based on the Diffuser model. The interaction between these two layers occurs during the generation phase. A high-level plan consisting solely of sub-goals is initially sampled at this stage. Following this, each successive pair of sub-goals is input into the low-level planner to create corresponding low-level plans. These plans are then compiled in sequence, according to the order of the sub-goals. Although, given this structure, there is no necessity for end-to-end training, it is an interesting suggestion to think further as a future work. This decoupling of high-level and low-level planning during the training phase enhances the efficiency of the training process.

---

> > > ### Comment · Reviewer_AgLN · 2023-11-22
> > > **Thanks for your detailed response!**
> > >
> > > Thank you for the clarifications, I've enjoyed the paper as well as the insightful discussions. It must have been a stretch running all these experiments in a short time. I have revised my rating to an 6!

---

### Official Review · Reviewer_u9ra · 2023-11-02

**Soundness:** 4 excellent
**Presentation:** 4 excellent
**Contribution:** 3 good
**Rating:** 6
**Confidence:** 4

**Summary:**

The paper proposes Hierarchical Diffuser (HD), a sub-goal based method for planning with diffusion models. In recent literature, Diffuser [1] was proposed to plan an optimal sequence of state and action between initial state and goal. Following diffuser, the authors design two diffusers. One diffuser plans for sub-goals between the start and the goal state. For each sub-goal segment, the second diffuser plans the optimal state-action sequence. Experiments are conducted on Maze, AntMaze, MuJoCo Gym and FrankKitchen benchmarks. The performance of HD is compared with relevant offline RL and hierarchical RL algorithms. Further, the experiments show improved receptibility, OOD generalization and less computation overhead as compared to diffuser. Relevant ablations are performed.

**Strengths:**

Discretizing planning into sub-goals with diffusion models is shown to be advantageous as more diverse scenarios can be solved. Low-level diffusion planning becomes task-agnostic.

Even the sparse diffuser or SD version is better than diffuser because of increased receptive field. The authors validate this by showing that increasing the kernel-size (hence the receptive field) of diffuser leads to better performance but weaker generalization.

SD with dense actions leads to better fitting of the sparse objective, a typical bottleneck in hierarchical RL.

HD takes lesser time than Diffuser due to shorter high-level sequence and parallelly solving low-level plans for all the segments.

**Weaknesses:**

The formulation has limited novelty. While a single diffuser is not sufficient for planning over long-horizons, the work introduces two diffusers: one sparse diffuser for planning sequence of sub-goals between initial state and goal, while a standard diffuser solves for individual sub-goal segments.

Given that there are also methods which perform state-only diffusion like Decision-diffuser [2], is it possible to perform relevant ablations to justify why states are not sufficient for having a good estimate of $J$? If you are following diffuser’s codebase, are you using the diffusion sampled actions or a separate controller for Maze2d tasks like diffuser?

Solving for sub-goals independently without the knowledge about the final goal might lead to non-optimal behavior. How do you ensure that the sampled path for one segment does not overlap with the paths for the other segment? How many trajectories do you sample for individual segments and for the sub-goal sequence?

Also, what happens when actions do not lead to the exact sub-goal (unless you use inverse dynamics actions) for the maze case? Because there is no-feedback from the low-level diffuser to high-level diffuser, it might become challenging.

Overall, I acknowledge that the presented method is a promising revision of Diffuser which performs better in all aspects. However, I believe that the contributions are not significant enough.

**Questions:**

See weakness above.

---

> ### Author Response · Authors · 2023-11-23
> **Response to Reviewer**
>
> ## R1: Limited novelty
>
> - Thank you for your comment regarding the perceived novelty of our formulation. While it is true that the core concept of using diffusion models for planning is not entirely new, the novelty of our work lies in the specific implementation and integration of these models in a hierarchical planning framework. The integration of a sparse diffuser with a standard diffuser is a novel aspect of our work. Also, our proposed method is very simple and thus practical compared to other method like HDMI. However, we show that this simple model achieves even better performance than the previous complex model.
>
> ## R2: Why states-only modeling (as in Decision Diffuser) are not sufficient for having a good estimation of $J$?
>
> - In our model, state-only modeling is not enough because we also skip intermediate states in our high-level planner. We think it is not a problem in Decision Diffuser because they do not skip intermediate states. Without knowing what has happened in the skipped intermediate states, the value estimation can be difficult.
>
> ## R3: Is sampled action or controller used in Maze2d tasks like Diffuser?
>
> - The same controller is used as in Diffuser on the Maze2D tasks for a fair comparison.
>
> ## R4: Solving sub-goals independently might lead to non-optimal behavior
>
> In the replanning mode of our method (the closed-loop mode), the low-level planner does not independently solve the segment because the high-level planner also updates the subgoal depending on the state of the low-level planner. In the open-loop mode, which we use when the task can be segmented more independently, we rely on the quality of the high-level planner to connect the low-level plans.
>
> ## R5: How is overlapping avoided during low-level sampling?
>
> The planing at the low-level is a goal-conditioned sub-task. The low-level planner takes in every two consecutive sub-goals in the high-level plan as starting state and goal state, and is trained to fill in intermediate states. Thus the low-level plans can be assembled according to the sub-goal order without overlapping.
>
> ## R6: How many trajectories are sampled for individual segments and sub-goal sequences?
>
> For goal-conditioned long-horizon tasks, one trajectory is sampled for both individual segments and sub-goal. For the Gym and Kitchen tasks, 64 trajectories are sampled.
>
> ## R7: What happens when sub-optimal action diverges from the sub-goal for the maze case?
>
> When sub-optimal action diverges from the sub-goal, resulting in an unexpected state, we correct the bad effect of sub-optimal actions by replanning. Increasing the replanning frequency improves the performance because the plan gets the chance to be corrected. It’s a trade-off between performance and computation resources. Other alternatives might also be worth exploring, i.e., explicitly learning a goal-conditioned policy $\pi_\theta(a|s, s_g)$ might be beneficial when the state and action space are high-dimensional, bringing challenges to modeling the whole sequence.
>
> ## R8: Contributions are not significant
>
> Thank you for your feedback regarding the perceived limited contribution of our work. We respectfully offer a different perspective, highlighting several contributions which has also been acknowledged by other reviewers:
>
> - **Improvements in Performance and Efficiency with a Much Simpler Model**: Our work with the HD framework has demonstrated notable improvements over our previous baselines in performance and computational efficiency. These advancements are substantial, as they contribute to our approach's effectiveness and practical applicability in real-world scenarios. This improvement has been recognized and underscored by reviewer AgLN. Our method is also much more simpler than the previous hierarchical diffusion method, HDMI.
> - **Theoretical Analysis and Experimental Validation**: We have proposed a novel approach and supported it with rigorous theoretical analysis and well-designed experiments. This comprehensive approach ensures that our contributions are grounded in theory and practice, providing a robust foundation for our findings. (reviewer AgLN)
> - **Insights into Receptive Field and Data Coverage**: Another significant aspect of our work, as highlighted by reviewer KFqH, is the identification of the relationship between the diffusion model’s receptive field and data coverage. This insight is particularly valuable as it opens new avenues for further research in the field.
>
> In summary, our contributions extend beyond mere incremental improvements. The combination of performance enhancements, theoretical analysis, and new insights into key aspects of diffusion models collectively represent significant advancements in the field. We are confident that our work offers valuable contributions and lays the groundwork for future research and development.

---

### Official Review · Reviewer_zGcK · 2023-11-03

**Soundness:** 3 good
**Presentation:** 3 good
**Contribution:** 2 fair
**Rating:** 5
**Confidence:** 5

**Summary:**

This paper proposed a modified version of the diffuser called Hierarchical Diffuser.  The Hierarchical Diffuser incorporates a two-tiered approach: a high-level "jumpy" planning strategy that can quickly survey a broader scope of possibilities with reduced computational demands, and a low-level planner that refines these broader plans into specific, actionable steps. This hierarchy allows the method to operate more efficiently and effectively, with the higher level providing guidance that simplifies the task for the lower level. Empirical evaluations show that this method outperforms both traditional diffusion-based planners and other hierarchical approaches in speed and performance on standard offline reinforcement learning benchmarks.

**Strengths:**

- Using a hierarchical structure makes sense in long-horizon planning.
- In replanning, a hierarchical structure is more efficient since it only needs to use low-level to
- Show the relationship between kernel size and generalization.

**Weaknesses:**

- Current SOTA diffusers seem to have a better performance. For example [1] have a 167 score on large maze2d.
- The Unet itself has a hierarchical structure. If an environment needs a hierarchical structure, a simple way is to increase the depth of the Unet. The authors might need to provide more results to show that they are better.
- The improvement in Mujoco is not enough for me.
- The paper said that they have evaluated generalization. However, the OOD task they test on is too simple. In other papers, harder OOD tasks are tested.  [1] add coins. [2] add obstacle.
- For hierarchical structure, it is important to ensure it is smooth on the connection points. However, the paper didn’t talk about it.

**Questions:**

I don’t understand the meaning of SD-DA. It seems like only improves the performance of walker2d. I feel like predicting so detailed actions loses the benefit of high-level planning.

---

> ### Author Response · Authors · 2023-11-13
> **Reference to papers [1] and [2]**
>
> Dear Reviewer zGcK,
>
> It seems that you missed providing the title of the papers [1] and [2]. Thank you!

---

> ### Comment · Reviewer_zGcK · 2023-11-13
> **comment**
>
> Sorry, my bad.
>
> [1] AdaptDiffuser: Diffusion Models as Adaptive Self-evolving Planners
>
> [2] Motion Planning Diffusion: Learning and Planning of Robot Motions with Diffusion Models

---

> ### Author Response · Authors · 2023-11-21
> **Response to Reviewer**
>
> ## R1: Current SOTA has a better performance, e.g. [1] 167 score on large mazed2d
>
> We respectfully disagree with the statement that AdaptDiffuser is considered the current state-of-the-art (SOTA) method superior to ours. **Comparing our method directly with AdaptDiffuser is unfair** due to several key factors.
> - First and foremost, it is unfair to compare our method with AdaptDiffuser [1] because **AdaptDiffuser relies on using privileged knowledge, specifically the true dynamics function $\mathcal{T}(s, a)$**. This strong assumption is usually not made in this line of research, which our method and the baselines are focusing on.
> - Secondly, it would not be appropriate to claim that AdaptDiffuser is superior to our method based only on the performance of a specific task. Instead, **one should consider the overall performance across multiple tasks rather than a single task.** In the DMC MuJoCo benchmark, **our method outperforms the AdaptDiffuser (despite it using the true transition dynamics) in 5 out of the 9 tasks**. Ours has a higher average score (84.6 for ours vs. 83.4 for AdaptDiffuser). Even in cases where AdaptDiffuser outperforms our method, the difference is small (e.g., 38.1 for ours vs. 38.3 for AdaptDiffuser in Med-Reply HalfCheetah). Additionally, it is worth noting that the **AdaptDiffuser paper does not evaluate the Multi2D tasks, which are considered a standard benchmark for diffuser-based planning models**. Therefore, it is unfair to conclude that AdaptDiffuser is generally the SOTA and better than ours based on the performance on large maze2D.
> - Lastly, we would like to emphasize that **the contributions of the two papers (ours and AdaptDiffuser) are orthogonal**. AdaptDiffuser introduces a data-augmentation approach, while our work introduces a hierarchical approach. These two ideas are not in competition but can be used in conjunction. We believe both ideas are valuable to the community, and there is no need to choose one. Exploring the synergy of these two ideas is an interesting avenue for future work.
>
> We hope this clarifies our position and highlights the unique contributions of our work. We appreciate the reviewer's feedback and the opportunity to address their concerns.
>
> [1] Zhixuan Liang, et al. AdaptDiffuser: Diffusion Models as Adaptive Self-evolving Planners.
>
> ## R2: Improvement in MuJoCo is not enough to me
>
> We deeply appreciate the valuable feedback provided by the reviewer. However, we respectfully disagree with the subjective judgment that there was "not enough" improvement in the MuJoCo benchmark. We would also like to kindly request further clarification on what "not enough" means more technically and scientifically.
>
> Nevertheless, we firmly believe that the improvement achieved in the MuJoCo benchmark is both meaningful and significant. **Our method consistently outperforms all baselines, with an average score of 84.6 compared to 77.5 for Diffuser and 82.6 for HDMI**. Although the improvement over HDMI may not appear substantial at first glance, it is crucial to note that this averaged score is calculated across 9 tasks, highlighting the *consistent performance enhancement of our approach*. Furthermore, we are proud to emphasize that our method achieves these results while maintaining a significantly simpler and faster framework, particularly compared to the HDMI baseline.
>
> Once again, we sincerely appreciate the reviewer's valuable feedback.
>
> ## R3: The Unet itself has hierarchical structure. Why not simply increase the depth of Unet? Provide more result on this.
>
> Thank you for the insightful feedback. We agree that this is an interesting observation and worth investigating. Following the suggestion, we conducted an additional experiment comparing a Diffuser with a deeper Multiscale U-Net (with twice as many layers) and HD. As shown in Table 1 below, we can see that HD significantly outperforms the deeper Multiscale U-Net. Furthermore, we found that the deeper Multiscale U-Net is computationally much more expensive, leading to very slow training and test times (Table 2 and Table 3 below). More details about the experiment can be found in the Appendix of the updated paper, which will be uploaded.
>
> Table 1: RL Performance Comparison on Maze2D
> |  Task | Multiscale U-Net | HD |
> | --- | :-: | :-: |
> | Umaze | 114.0 +- 4.6 | **128.4 +- 3.6** |
> | Medium | 117.8 +- 1.9 | **135.6 +- 3.0** |
> | Large | 124.8 +- 6.0 | **155.8 +- 2.5** |
>
> Table 2: Training Wall-Clock Time (sec.) Comparison on Maze2D
> | Task | Multi-Scale U-Net | HD |
> | --- | :-: | :-: |
> | Umaze | 29.6 | **8.0** |
> | Medium | 137.4 | **8.7** |
> | Large | 141.8 | **8.6** |
>
> Table 3: Testing Wall-Clock Time (sec.) Comparison on Maze2D
> | Task | Multi-Scale U-Net | HD |
> | --- | :-: | :-: |
> | Umaze | 1.6 | **0.8** |
> | Medium  | 10.4 | **3.1** |
> | Large  | 10.4 | **3.3** |

---

> ### Author Response · Authors · 2023-11-21
> **Continued - Response to Reviewer**
>
> ## R4: The OOD task is too simple. Harder OOD tasks are tested in other papers. [1] add coins [2] add obstacle
>
> Thank you for the feedback. Based on your suggestion, we conducted an additional out-of-distribution (OOD) experiment on one of the suggested tasks, specifically the obstacle task used in the MPD model [1]. Following the methodology of MPD [1], we applied our HD model to the PointMass2D Dense task with OOD obstacles. To do this, we replaced the flat Diffuser planner used in MPD with our HD model, referred to as MP-HD. We used the official code of MPD for comparison.
>
> As shown in Table 4 below, our HD model performs significantly better than MPD on these challenging OOD tasks. We also visualized sample trajectories of HD from two randomly selected (start, goal) pairs [here](https://anonymous.4open.science/api/repo/plan_with_HD-E821/file/pointmaze2d-ood.png). These trajectories demonstrate that our model can effectively avoid collisions when faced with OOD test obstacles.
>
> (Note that the official MPD code yielded a success rate of 75.0 ± 43.3, which is lower than the performance reported in the MPD paper (79.0 ± 40.7). In a personal conversation with the author, we were informed that reproducing the score in the paper using the official code is currently challenging for some reason.)
>
> Table 4: RL Performance Comparison on PointMass2D Dense - Extra Obstacles
> | Method | Success Rate | Training Time (sec., 100 Updates) | Testing Time(sec., 50 Samples) |
> | --- | :-: | :-: | :-: |
> | MP-Diffusion | 75.0 +- 43.3 | 5.1 | 0.62 |
> | MP-HD (Ours) | **81.0 +- 38.8** | **3.8** | **0.59** |
>
> [1]: Joao Carvalho, et al. Motion Planning Diffusion: Learning and Planning of Robot Motions with Diffusion Models
>
> ## R5: It is important to ensure smoothness on the connection points. The paper didn’t talk about it.
>
> Thank you for the constructive feedback. While we understand the significance of this aspect, we would like to clarify that our method has been empirically observed to exhibit robustness in terms of smoothness at the connection points because, in our experiments, we have achieved high performance without introducing any specific means to address this issue.
>
> In our approach, the inpaint (open-loop) mode ensures smoothness as we perform inpainting to fill the states between the subgoals suggested by the high-level planner. This creates a natural flow, where a subgoal becomes the starting state for the next subgoal, resulting in smoothness at the connection points. Additionally, in the replanning (closed-loop) mode, the high-level planner continuously replans the subgoals at each time step, eliminating the need to be concerned about smoothness at the connection points.
>
> We appreciate the reviewer's feedback on this topic, and based on their input, we will further discuss and elaborate on this aspect in the revised version of our paper.
>
> ## Q1: I don’t understand the meaning of SD-DA. It seems like only improves the performance of walker2d. I feel like predicting so detailed actions loses the benefit of high-level planning.
>
> SD-DA is a model for ablation study to help understand what aspects of our final model contribute to the performance. The meaning of this model is written in detail in the paper as follows.
>
> > The missing states and actions in the subsampled trajectories $x^\text{SD}$ might pose difficulties in accurately predicting returns in certain cases. Therefore, we investigate a potential model improvement that subsamples trajectories with sparse states and dense actions. The hypothesis is that the dense actions can implicitly provide information about what has occurred in the intermediate states, thereby facilitating return prediction.
> >
>
> Table 4 in the paper shows that including dense action significantly improves value estimation performance. Additionally, it contributes to RL performance, although the degree of improvement varies across different tasks. Notably, walker2d shows the most significant improvement, while the effect on other tasks is less pronounced.
>
> In conclusion, these experiments serve as an ablation study to gain a better understanding of the model. We believe that such analysis provides crucial knowledge to the community.

---

> > ### Author Response · Authors · 2023-11-23
> >
> > We kindly request the reviewer to provide us with the response and let us know whether your concerns have been addressed, as the deadline for the discussion phase is fast approaching.

---

> ### Comment · Reviewer_zGcK · 2023-11-30
> **Official comment**
>
> R1: Thanks for clarifying the two papers. To me, the transition function is not hard to get (or train) in a mazed2d. I believe a better way to show your method is to combine this two together and get a even better SOTA.
>
> R2: 84.6 and 82.6 is really just not too impressive.
>
> R3: Thanks for the experiment. It address some of my concerns.
>
> R4: Thanks for the experiment. It address my concerns. It will be great if you also have the result for the coins.
>
> Q1: Since it can only improve one of third environments, it is not that clear if it is something that can improve the method consistently.
>
> I have raised the rating from 3 to 5 due to R1, R3, and R4.

---

### Meta-Review · Area_Chair_hfSz · 2023-12-20

**Metareview:**

The paper presents the Hierarchical Diffuser, a two-level planning method based on diffusion models. Reviewers appreciated original integration of existing components, simplicity, demonstrated effectiveness and analysis, the paper's clarity of exposition and additional experiments conducted during rebuttal period. Concerns were raised about the novelty of the method compared to existing approaches, and limited improvement in certain environments. Nonetheless, the reviewers agreed the work presents a valuable contribution to the community and therefore I recommend acceptance.

**Justification For Why Not Higher Score:**

Limited novelty compared to existing methods.

**Justification For Why Not Lower Score:**

Simplicity.

---

### Decision · Program_Chairs · 2024-01-16

Accept (poster)